# Structures of TGF-β with betaglycan and signaling receptors reveal mechanisms of complex assembly and signaling

Łukasz Wieteska [1,2,3], Alexander B. Taylor [4], Emma Punch[5], Jonathan A. Coleman [1], Isabella O. Conway[1], Yeu-Farn Lin [6], Chang-Hyeock Byeon [1], Cynthia S. Hinck[1], Troy Krzysiak[1], Rieko Ishima[1], Fernando López-Casillas [7], Peter Cherepanov [5], Daniel J. Bernard [6], Caroline S. Hill [3] ✉ & Andrew P. Hinck [1] ✉

Betaglycan (BG) is a transmembrane co-receptor of the transforming growth factor-β (TGF-β) family of signaling ligands. It is essential for embryonic development, tissue homeostasis and fertility in adults. It functions by enabling binding of the three TGF-β isoforms to their signaling receptors and is additionally required for inhibin A (InhA) activity. Despite its requirement for the functions of TGF-βs and InhA in vivo, structural information explaining BG ligand selectivity and its mechanism of action is lacking. Here, we determine the structure of TGF-β bound both to BG and the signaling receptors, TGFBR1 and TGFBR2. We identify key regions responsible for ligand engagement, which has revealed binding interfaces that differ from those described for the closely related co-receptor of the TGF-β family, endoglin, thus demonstrating remarkable evolutionary adaptation to enable ligand selectivity. Finally, we provide a structural explanation for the hand-off mechanism underlying TGF-β signal potentiation.

Secreted ligands of the Transforming Growth Factor-β (TGF-β) family transduce their signals by binding and bringing together type I and type II receptor serine/threonine kinases, which triggers a transphosphorylation cascade and activation of downstream signaling[1]. Similar to other cytokine families, some ligands of the TGF-β family bind co-receptors, which govern receptor binding and signaling[2]. One such co-receptor is the cell surface proteoglycan TGFBR3, also known as betaglycan (BG). Unlike the type I and type II receptors, BG lacks a cytosolic kinase domain and is not directly involved in transducing the signal[3]. BG is nonetheless essential in vivo, where it binds and potentiates the activity of the three TGF-β isoforms (TGF-β1, -β2 and

-β3) by presenting them to their signaling receptors, TGF-β receptor type I (TGFBR1) and TGF-β receptor type II (TGFBR2)[4]. Once the ligand–receptor signaling complex is formed in a stepwise manner, first by binding TGFBR2 and then by recruiting TGFBR1, TGFBR2 phosphorylates the co-complexed TGFBR1, which phosphorylates and activates SMAD2 and SMAD3. TGFBR1 can also transphosphorylate other type I receptors, such as ACVR1, activating the non-canonical SMAD1/5/9 pathway[5].

BG binds the three TGF-β isoforms, 25 kDa disulfide-linked homodimers[6] with comparable high affinity ($K_D$ 5–20 nM)[7]. BG is most critical for potentiation of TGF-β2 signaling as this ligand binds

[1]Department of Structural Biology, University of Pittsburgh, Pittsburgh, PA, USA. [2]Faculty of Biological Sciences, Astbury Centre for Structural Studies, University of Leeds, Leeds, UK. [3]Developmental Signalling Laboratory, The Francis Crick Institute, London NW1 1AT, UK. [4]Department of Biochemistry & Structural Biology and Greehey Children's Cancer Research Institute, The University of Texas Health Science Center at San Antonio, San Antonio, TX, USA. [5]Chromatin Structure and Mobile DNA Laboratory, The Francis Crick Institute, London NW1 1AT, UK. [6]Department of Pharmacology and Therapeutics, McGill University, Montreal, QC, Canada. [7]Departmento de Biología Celular y del Desarrollo, Instituto de Fisiología Celular, UNAM, Mexico City, Mexico. ✉e-mail: caroline.hill@crick.ac.uk; ahinck@pitt.edu

TGFBR2 with 200–500-fold lower affinity than TGF-β1 and -β3. In the absence of BG, cells are largely unresponsive to TGF-β2, but in its presence, cells respond to concentrations comparable to the other isoforms[4,8,9]. BG-null mice die during embryonic development and share many of the phenotypic characteristics of the TGF-β2 knockout, including severe heart and liver defects, consistent with the requirement of BG for TGF-β2 signaling[10]. BG is nonetheless important for the function of all three TGF-β isoforms, by sensitizing non-transformed cells to TGF-β-mediated growth inhibition[8,10]. Downregulation of BG by transcriptional repression or ectodomain shedding, a process by which BG is proteolytically cleaved and the soluble ectodomain is released into the extracellular space where it can bind and sequester TGF-βs from the signaling receptors[9], correlates with oncogenic transformation in cancer cells[11–13]. Hence, BG is regarded as contributing to the tumor suppressive activity of the TGF-βs in vivo[9,14,15].

In vivo, BG is required in gonadotrope cells for the activity of inhibin A (InhA), a TGF-β family heterodimer that blocks follicle stimulating hormone (FSH) secretion from the anterior pituitary[16]. InhA does this by forming a high-affinity complex that includes both BG and an activin type II receptor, ACVR2A or ACVR2B[17], thereby sequestering the type II receptors from activins and other ligands which use them to activate signaling and subsequently induce expression of the FSH-β subunit[18].

BG, and a homologous co-receptor of the TGF-β family known as endoglin (ENG) which potentiates the function of BMP9 and BMP10, are transmembrane proteins with a large (approximately 760 amino acids) extracellular region comprised of two domains: the membrane-distal orphan domain and the juxtamembrane zona pellucida (ZP) domain[19,20] (Fig. 1a). The orphan domain can be divided into two subdomains, OD1 and OD2, yet these are closely connected through paired antiparallel β-strands and adopt a well-defined orientation relative to one another[21]. The BG and ENG orphan domains (BG$_O$ and ENG$_O$, respectively) bind the TGF-βs and BMP9/10, respectively, though structural differences within the BMP9/10 binding interface between these domains suggest that TGF-β binding by BG$_O$ is unlikely to mirror BMP9 binding by ENG$_O$[21,22]. The ZP domain can also be subdivided into two subdomains, the N-terminal ZP-N and the C-terminal ZP-C. ZP-N is modified by two glycosaminoglycan (GAG) chains, though it does not contribute to ligand binding either directly or through its GAG chains in BG or ENG[23,24]. The BG ZP-C domain (BG$_{ZP-C}$) binds the three TGF-β isoforms and InhA[25], while ENG ZP-C does not contribute to ligand binding and instead scaffolds the orphan domain for ligand binding. ZP-C is structurally similar to other ZP domains, including those in uromodulin, ZP3, and DMBT[26]. BG$_{ZP-C}$ is also homologous to TGFBR3L, another co-receptor of the TGF-β family[27] which serves to potentiate the inhibitory activity of inhibin B (InhB)[27,28]. BG binds TGF-β homodimers with 1:1 stoichiometry, engaging with both the orphan and ZP-C domains[4,7]. However, the isolated ZP-C domain binds TGF-β homodimers with 2:1 stoichiometry, effectively competing with TGFBR2 for binding[7]. In contrast, BG binds the α subunit of InhA via only its ZP-C domain[19,29–31] forming a 1:1 complex[7,29]. Mutagenesis and nuclear magnetic resonance (NMR) shift perturbation mapping of TGF-β and InhA are mostly in agreement, placing the interacting residues on the underside of the fingers, two β-hairpins that extend from the cystine knot core of each monomer[25,32]. However, reports on how BG$_{ZP-C}$ engages the ligands are contradictory[29,33,34].

The uncertainty of how BG binds TGF-βs and InhA to potentiate their function is due to the absence of structural information for the complexes. This, together with the pivotal role that BG has on modulating the activities of TGF-βs and InhA, whose dysregulation can drive cancer progression[35] and impact fertility[27] respectively, motivated us to determine the structure of BG to TGF-β with the signaling receptors, which we have accomplished using X-ray crystallography and Cryo-EM. We also developed a competition assay in living cells, which, along with mutational analysis and predictions from Alphafold2 Multimer (AF2M)[36], allowed us to identify the key determinants of ligand specificity. Together, this study provides a detailed molecular-level understanding of how BG achieves ligand selection and potentiation of signaling, which involves sequential binding of the type II and type I signaling receptors, concomitant with stepwise displacement of the co-receptor[7,37].

## Results

### Structure of the TGF-β:BG complex

Initial trials to crystallize TGF-β2 bound to full-length BG complex presented difficulties, most likely due to the disordered 85-residue linker connecting the orphan and ZP domains. It was also not possible to determine the structure of this complex using Cryo-EM due to sample heterogeneity and strong preferential orientation. We therefore turned to a 'divide-and-conquer' approach and attempted to crystallize the TGF-β2:(BG$_{ZP-C}$)$_2$ and TGF-β2:BG$_O$ complexes separately. The isolated TGF-β2:(BG$_{ZP-C}$)$_2$ complex again could not be crystalized, so we turned to a previously engineered mini monomer of TGF-β2, mmTGF-β2 (ref. 38). In spite of being a monomer and lacking the heel helix (Fig. 1b), mmTGF-β2 is much more soluble than TGF-β dimers and retains binding to BG$_{ZP-C}$[25]. This approach yielded high-quality orthorhombic crystals, enabling determination of the structure of the mmTGF-β2:BG$_{ZP-C}$ complex at 1.9 Å resolution using X-ray crystallography (Fig. 1c, Supplementary Fig. 1a, Supplementary Table 1).

Unlike BG$_{ZP-C}$, BG$_O$ does not prevent recruitment of TGFBR2 to TGF-βs, thus to understand the potentiation mechanism, we wanted to gain structural information on how the BG:TGF-β complex engages with the signaling receptors. We therefore attempted to crystallize the TGF-β3:BG$_O$:(TGFBR2)$_2$ complex. We used TGF-β3 instead of TGF-β2 to ensure high affinity binding of TGFBR2 and thus a stoichiometric complex. We also used zebrafish BG$_O$, designated hereafter as zfBG$_O$, rather than rat BG$_O$ (ratBG$_O$) owing to previous success crystallizing the former but not the latter[21]. While crystallization of this complex was not successful, we were able to obtain a low-resolution map of this complex using Cryo-EM and tentatively placed all components (Fig. 1d, e). We were not able to obtain a high resolution reconstruction due to streaky elongations in the density resulting from preferred orientations of the particles[39] and to the expanded densities in the areas attributed to TGFBR2 (ref. 39). We hypothesized that the expanded density, as well as the difficulty obtaining crystals with this complex, was a result of the previously characterized closed-to-open transition of TGF-β3, which results in a large scale reordering of the TGF-β protomers relative to one another (Fig. 1f)[40].

We therefore sought to stabilize the TGF-β dimer in a single conformation and prepared an alternative complex replacing TGF-β3 with TGF-β1 inTGF-β1:zfBG$_O$:(TGFBR2)$_2$ (Fig. 1g). We expected that the strong propensity of TGF-β1 to adopt the closed form[40] would reduce or completely diminish the conformational heterogeneity. Additionally, in a separate approach we wanted to obtain the structure of the TGF-β:BG$_O$ complex bound to both of the signaling receptors, TGFBR1 and TGFBR2. As the TGF-β1:(TGFBR2)$_2$:(TGFBR1)$_2$ heterotetrameric assembly cannot recruit BG$_O$, we aimed to construct a variant with the signaling receptors bound to only one monomer of TGF-β. We therefore switched to the TGF-β3 heterodimer, TGF-β3WD, a mutated form that indeed can bind only one pair of TGFBR2:TGFBR1[41]. We successfully assembled the TGFβ3WD:zfBG$_O$:TGFBR2:TGFBR1 complex (Fig. 1j) and in this case we used the more flexible TGF-β3, expecting that the bound TGFBR1, which spans the TGF-β dimer interface, would lock TGF-β3 in the closed conformation[42]. Cryo-EM analysis of both complexes yielded density that closely accommodated the component structures and enabled domain placement without ambiguity (Fig. 1h, i, Supplementary Fig. 2a, b and Supplementary Table 2). Despite this, it was difficult to build molecular models due to strong preferential alignment of the particles (Supplementary Fig. 3a–d). Thus, we returned to crystallography and successfully obtained

 

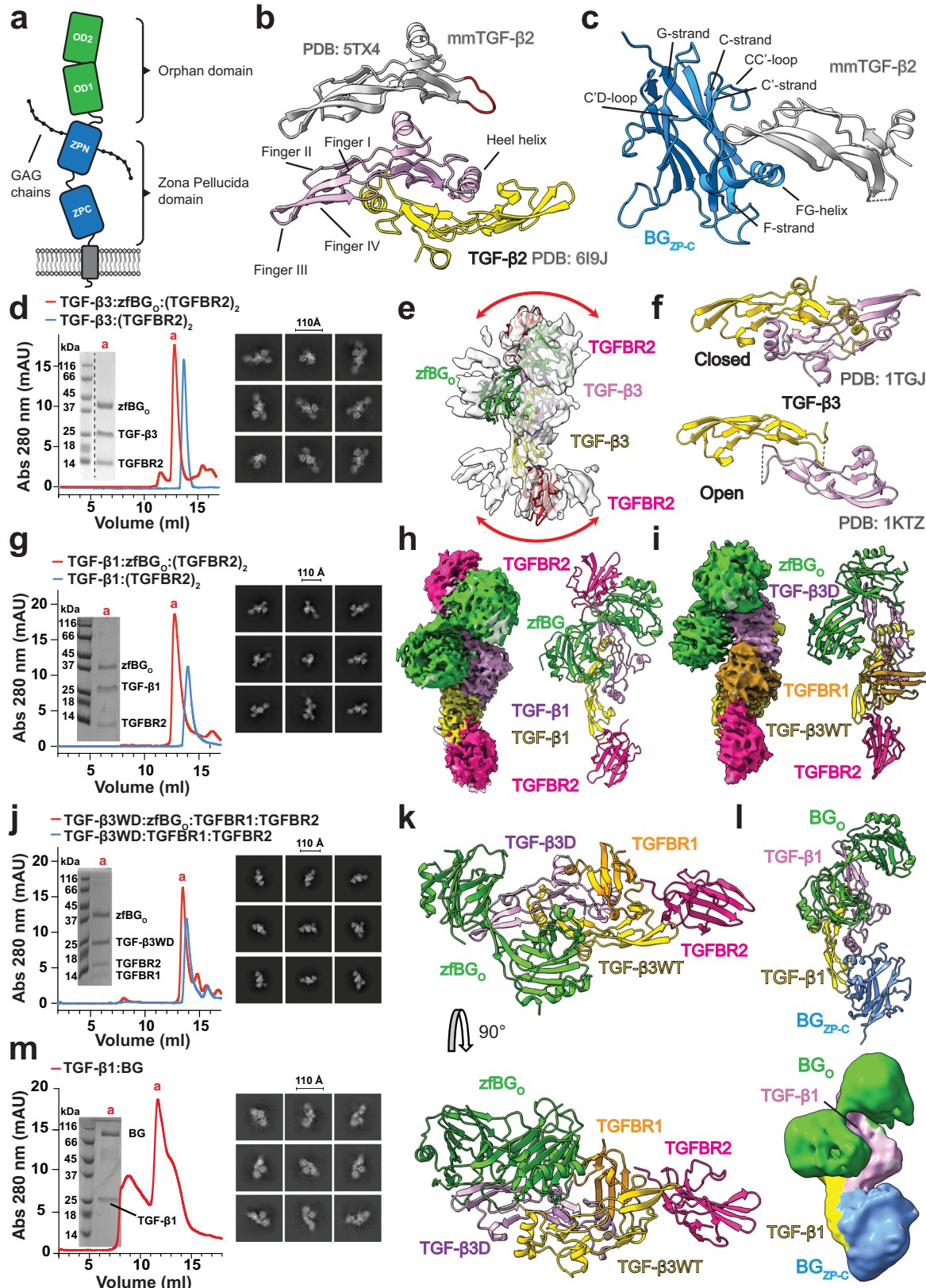

orthorhombic crystals of the TGF-β3WD:zfBG_O:TGFBR2:TGFBR1 complex and determined the structure at a resolution of 3.0 Å (Fig. 1k, Supplementary Fig. 1b, Supplementary Table 1).

With knowledge of the substructures of the two component domains of BG bound to TGF-β, we repeated the analysis of full-length rat BG in complex with TGF-β, using TGF-β1 instead of TGF-β2 (Fig. 1m).

We successfully derived a Cryo-EM map of the entire complex, albeit at low resolution, into which we were able to unambiguously position the BG_ZP-C and BG_O domains onto the TGF-β dimer, which agreed with the positioning obtained from the substructures (Fig. 1l, Supplementary Fig. 3e–g, Supplementary Fig. 2d, Supplementary Table 2). Further, the full-length structure revealed no additional density corresponding to

**Fig. 1 | X-ray crystallographic and Cryo-EM structures of BG$_{ZP-C}$ and zfBG$_O$ in complex with TGF-β. a** Schematic illustrating the BG domain organization. Both the orphan domain (BG$_O$) and the C-terminal portion of the zona pellucida domain (BG$_{ZP-C}$) are engaged in mediating TGF-β binding. **b** Comparison between previously determined structures of TGF-β2 dimer (PDB: 6I9J) and mmTGF-β2 (PDB: 5TX4). The loop in the latter, which replaces the heel helix in the dimer, is highlighted in red. **c** Visualization of the X-ray crystallographic structure of mmTGF-β2 bound to BG$_{ZP-C}$ (PDB: 8DC0) with highlighted structural features of the BG$_{ZP-C}$ domain. Size exclusion chromatography (SEC) purification profiles demonstrating distinct peaks of assembled complexes of TGF-β3:zfBG$_O$:(TGFBR2)$_2$ (**d**); TGF-β1:zfBG$_O$:(TGFBR2)$_2$ (**g**); TGF-β3WD:zfBG$_O$:TGFBR1:TGFBR2 (**j**) and TGF-β1:ratBG (**m**), as assessed by SDS-PAGE (insets). Full complexes are highlighted in red (peaks "a"), whereas partial complexes are highlighted in blue. Corresponding 2D classes provide an overview of the dataset quality and coverage of different views and data redundancy. Scale bar, 110 Å. **e** Low resolution Cryo-EM map of the TGF-β3:zfBG$_O$:(TGFBR2)$_2$ complex. **f** Open and closed conformations of TGF-β3 that had been previously captured by x-ray crystallography (PDB: 1KTZ and 1TGJ). **h, i** Respective Cryo-EM maps of imaged complexes (**g** and **j**) with structural models and highlighted, color-coded components (PDB: 9FKP, 9FK5, respectively). **k** Visualization of the X-ray crystallographic structure of the TGF-β3WD:zfBG$_O$:TGFBR1:TGFBR2 complex (PDB: 9B9F). **l** Low resolution Cryo-EM map of TGF-β1:ratBG complex (EMD-50326) with highlighted components. Source data are provided as a Source Data file.

BG$_{ZP-N}$, suggesting that the positioning of this domain is not fixed relative to the BG$_{ZP-C}$. We also confirmed that BG$_O$ and BG$_{ZP-C}$ are placed on the opposite monomers of TGF-β, without any contact between the domains (Fig. 1l, Supplementary Fig. 3g).

In addition to our experimental pursuits, we explored the potential of AF2M[36] to predict the structures of these complexes. While AF2M capabilities were challenged when full-length BG complexed with TGF-β was modeled, the predictions were impressively accurate for the individual domains, BG$_O$ and BG$_{ZP-C}$ complexed with TGF-β (Supplementary Fig. 3h, I; Supplementary Data 1 and 2). This demonstrates the utility of this recent technology for simulating the protein–protein interactions we aim to visualize. We further leveraged this capability to gain a better understanding of substrate specificity, as described below.

### Betaglycan ligand specificity

Among the 33 members of the TGF-β family, BG is known to bind with high affinity to the three TGF-β isoforms through both the BG$_{ZP-C}$ and BG$_O$ domains[19,43] and to the α subunit of InhA through the BG$_{ZP-C}$ domain[27]. It was proposed that BG can also bind to various BMPs[44,45], but biophysical measurements contradicted these claims[25]. To determine the BG:ligand selectivity, we developed a competition assay to test if selected ligands of the TGF-β family can compete with TGF-β2 for binding to BG. For this, we established a HEK293T cell line stably overexpressing rat BG with an N-terminally fused SNAP-tag (Supplementary Fig. 4a, c) that allows for specific labelling through click chemistry[46]. After confirming that the SNAP tag did not negatively impact BG activity (Supplementary Fig. 4b) and that fluorescently labeled SNAP-BG signal colocalized with fluorescently labeled TGF-β2 (Fig. 2a and Supplementary Fig. 4d), we assessed the ability of 100-fold excess of unlabeled ligand to compete with the labeled TGF-β2. Labeled TGF-β2 bound to SNAP-BG could be efficiently outcompeted by unlabeled TGF-β1, TGF-β2 and TGF-β3, but ActA, BMP2, and BMP4 were not able to compete (Fig. 2a, b). We further demonstrated that the co-localization of fluorescently labeled BG and TGF-β could be decreased by unlabeled TGF-βs in a dose-dependent manner (Supplementary Fig. 4e). Together, these data suggest that among the ligands tested, only TGF-β isoforms can bind BG with high affinity.

Building upon the success of our structural predictions using AF2M, we expanded our survey by integrating AF2M modeling to encompass a broader spectrum of growth factors within the TGF-β family. The ptm+iptm score computed for BG$_O$ or BG$_{ZP-C}$ in complex with the TGF-βs, which provides a probability estimate assessing both post-translational modifications (PTMs) and inter-residue contacts (iPTMs) within the protein structures, was significantly higher than for other ligands, suggesting that BG is much more likely to bind TGF-β isoforms compared to any other TGF-β family ligand (Fig. 2c, d), which agrees well with our competition assay. As expected, predictions for InhA binding to BG$_{ZP-C}$ also produced an elevated ptm+iptm score compared to other ligands, in agreement with the experimental data (see below). In addition, the lower score for InhA compared to TGF-β isoforms likely reflects lower binding affinity compared to TGF-βs[27].

However, it should be noted that the AF2M predictions for BG$_{ZP-C}$ span a narrower range between true positive and negative cases compared with those for BG$_O$, suggesting the need for caution in interpreting the BG$_{ZP-C}$ results.

### Interactions between TGF-β and the BG ZP-C domain

The structure presented in Fig. 1c shows that BG$_{ZP-C}$ uses the surface of the exposed sheet formed by the C, C', F, and G strands to pack against the fingertips and the inner surface of fingers I-II and III-IV of mmTGF-β2. The BG$_{ZP-C}$ binding site on TGF-β lies on the inner surface of the fingers of TGF-β2, involving residues L28, I33, V61, L86, I88, I92, K97, E99, and L101. It was previously suggested that I92, K97, and E99, which are essential for binding BG$_{ZP-C}$, are conserved among the TGF-β isoforms and inhibin α subunit, setting these ligands apart from other members of the family[25]. In the crystal structure of the mmTGF-β2:BG$_{ZP-C}$ complex, V92 (I92 in wild type TGF-β2), K97, and E99, which lie within fingers III and IV, indeed form key contacts with BG$_{ZP-C}$ (Fig. 3a I–III). In addition, the concave surface of TGF-β finger IV forms hydrophilic interactions with the irregularly structured region of the FG loop, particularly mmTGF-β2 residues K97, Y90, and E99, which interact with BG$_{ZP-C}$ residues L156, D157, A158, and T159, respectively. It is notable that most of these interactions are not formed between side chains, but rather between the side chain of one residue and the backbone amide or carbonyl of another residue. The tips of the fingers of the ligand are also engaged and form contacts with residues located on the CC'- (P71, D72) and C'D-loop (N82, K86, D88), as well as the C' strand (E81), with N82 being vital as assessed by Surface Plasmon Resonance (SPR) (Fig. 3b, c). In addition, E130, located on the F strand of the BG$_{ZP-C}$ domain, participates in a water-mediated hydrogen bond with the tip of the fingers I and II.

In earlier structures of rat and mouse BG$_{ZP-C}$ domain alone, the FG loop, which bridges the penultimate and ultimate β-strands (F and G, respectively), had weak electron density and was either modeled in an extended geometry[33] or not at all[34]. In contrast, in the structure of the mmTGF-β2:BG$_{ZP-C}$ complex, the FG loop undergoes a disorder-to-order transition and forms a helix that also packs against the inner surface of fingers I-II and III-IV of mmTGF-β2 (Fig. 3a III). Hydrophobic residues A158, I161, W162, and M165 form crucial interactions with the concave surface of the growth factor fingers, including W30, W32, V92 and L101 (Fig. 3a III) and mutations within this region severely disrupted binding (Fig. 3b).

It was reported that the affinity of mmTGF-β2 for binding BG$_{ZP-C}$ is weakened by 10-fold compared to TGF-β2 (ref. 25). However, an overlay of mmTGF-β2 with a full TGF-β2 monomer did not suggest any additional contacts that could account for the affinity difference. Hence, we considered the possibility of additional interactions between BG$_{ZP-C}$ and the opposing monomer of TGF-β that are absent in mmTGF-β2:BG$_{ZP-C}$. Indeed, alignment of the mmTGF-β2 as bound to BG$_{ZP-C}$ onto the TGF-β2 homodimer suggested that the FG-helix and heel helix/pre-helix extension of the second monomer might contact one another. This extends the interface in this area and is consistent with the positioning of BG$_{ZP-C}$ in the TGF-β1:BG structure (Fig. 1l) and

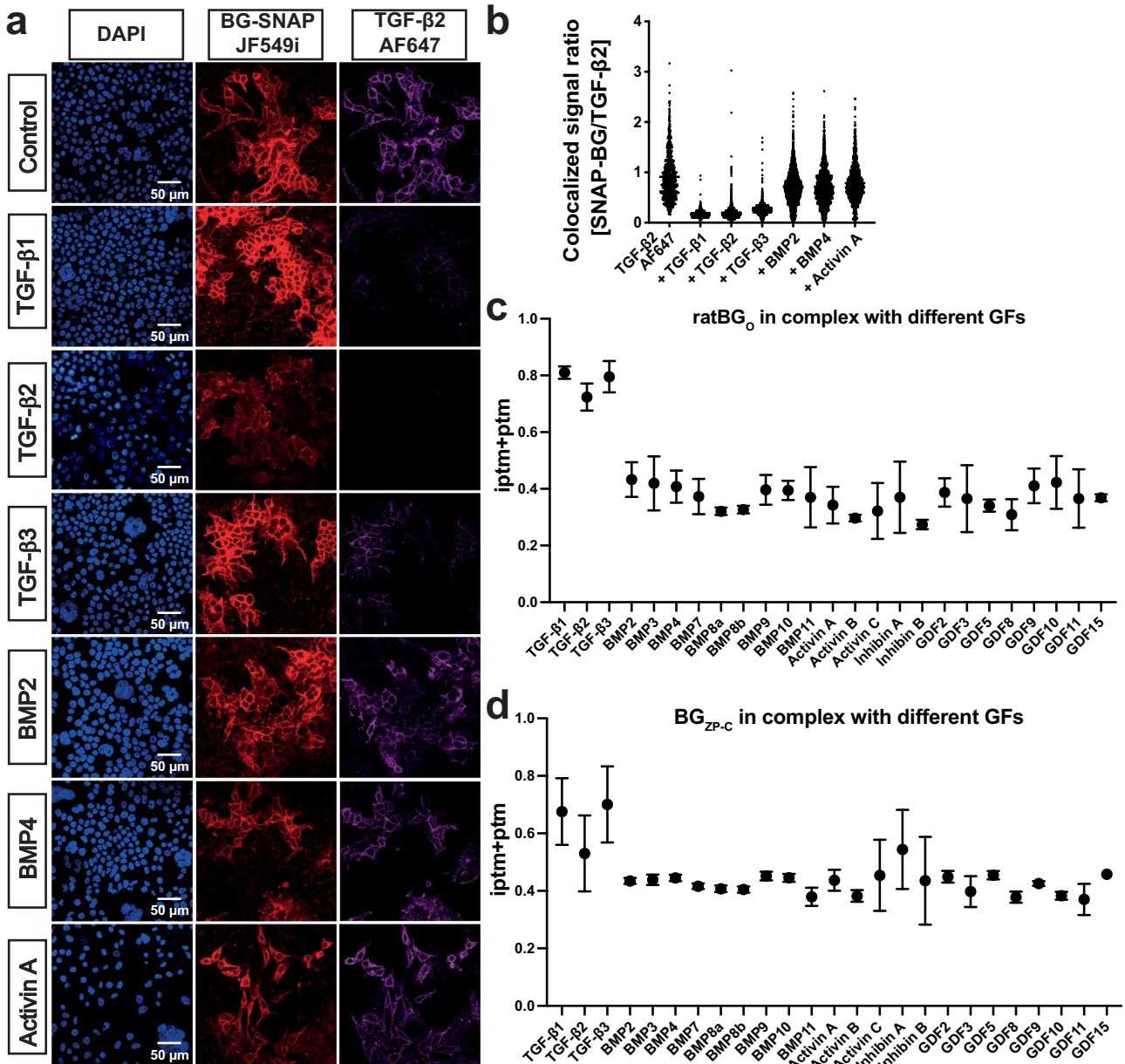

**Fig. 2 | Selectivity of BG towards TGF-β family ligands. a** Confocal microscopy images of HEK293T cells stably transfected with a SNAP-tagged BG construct (SNAP-BG) stained with DAPI (left panels, blue), incubated with SNAP-tag ligand conjugated to the CF567 fluorescent dye (middle panels, red), along with TGF-β2 tagged with AF647 fluorescent dye (right panels, violet) alone (row 1) or with the addition of an excess of indicated unlabeled ligands (rows 2–7). Scale bar, 50 μm. **b** Quantification of the fluorescent colocalized SNAP-BG/TGF-β2 microscopy data. Each data point represents the ratio of SNAP-BG signal to TGF-β2 signal within a

segmented region of the image generated by applying the Otsu thresholding method to the SNAP-BG channel. Analysis was performed on three images per condition, each corresponding to an independent biological replicate (*n* = 3). AF2M analysis for ratBG$_O$ (**c**) and BG$_{ZP-C}$ (**d**) complex formation capability with selected ligands. The y-axis represents the iptm+ptm score mean values with standard deviation derived from a total of 25 models generated for each complex. GF, growth factor. Source data are provided as a Source Data file.

an AF2M model of the TGF-β2:BG$_{ZP-C}$ complex (Fig. 3d). To investigate this, we simultaneously mutated Y50 and L51 or Q57, K60 and L64 in the pre-helix extension or helix region of TGF-β2 where the conserved residues could engage BG$_{ZP-C}$. However, both mutants displayed only marginally reduced binding affinity towards the BG$_{ZP-C}$, indicating other residues contribute to binding, or additional interactions are formed between side chains and the backbone in the proposed interface (Fig. 3e and Supplementary Fig. 5a). Interestingly, mutations of Q57, K60 and L64 significantly impacted binding of zfBG$_O$ (see below).

To investigate the functional impact on TGF-β signaling potentiation of the key residues identified above, we co-transfected L6E9 rat

muscle myoblasts, which express little to no BG, with BG$_{ZP-C}$ constructs with N82 or I161 substituted with alanine, together with a TGF-β responsive CAGA$_{12}$-luciferase reporter construct[47]. Upon stimulation of the transfected cells with low concentrations of TGF-β2, mutation of both N82 and I161 markedly diminished CAGA$_{12}$-luciferase activity in comparison to wildtype (WT) BG$_{ZP-C}$ (Fig. 3f). These findings confirm that binding-deficient BG$_{ZP-C}$ mutants diminish TGF-β signal potentiation.

To further confirm that the BG$_{ZP-C}$ mutations primarily influence ligand binding, without disrupting the structure, we recorded NMR natural abundance 2D $^1$H-$^{13}$C correlation spectra (Supplementary

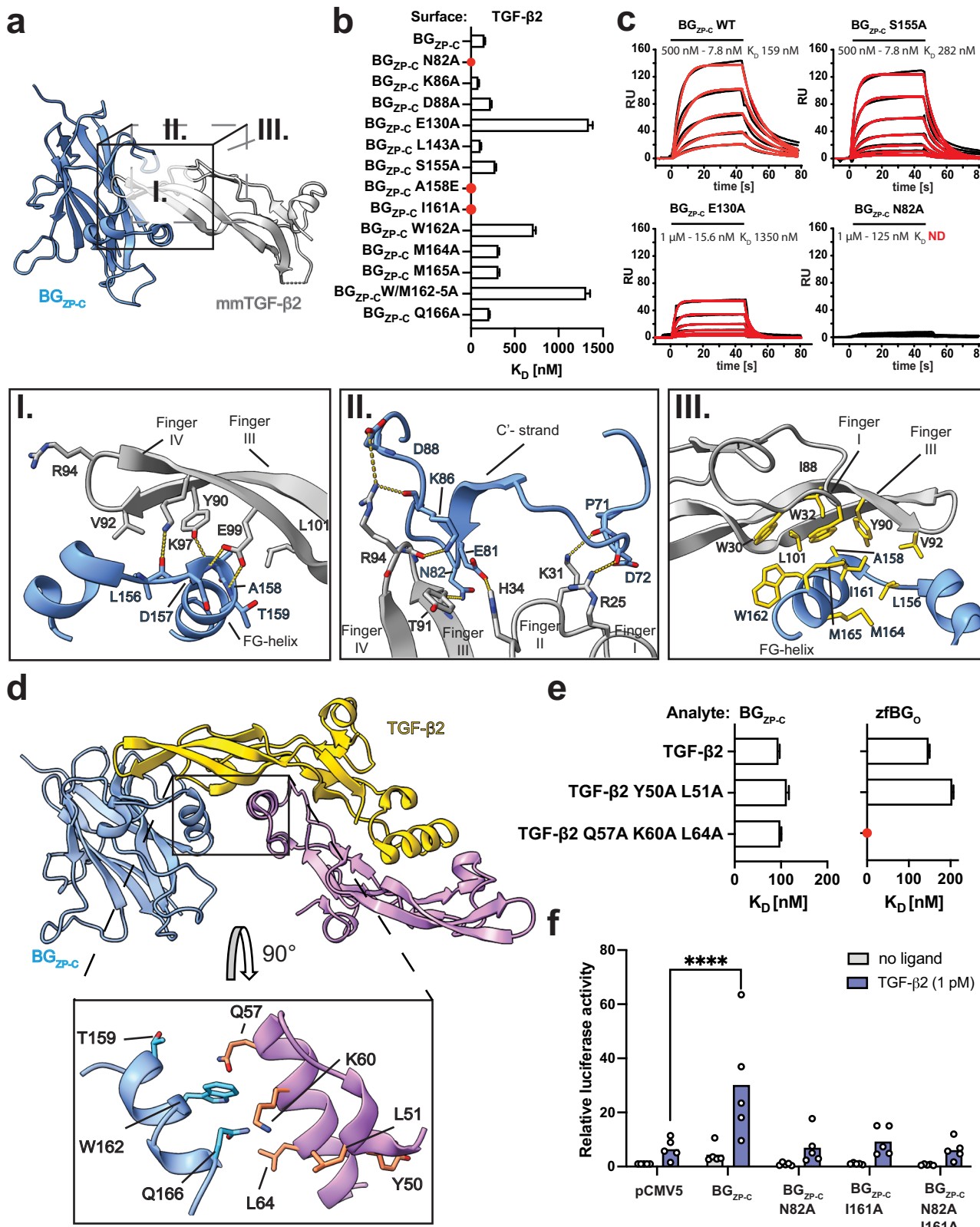

Fig. 5b–e). Comparison of the spectral fingerprints of the mutant $BG_{ZP-C}$ proteins against those of WT $BG_{ZP-C}$ confirmed that the mutations did not introduce significant global changes to the structures. This supports the conclusion that N82 and I161 of $BG_{ZP-C}$ play critical roles in TGF-β signaling by specifically binding TGF-β ligands.

## Interaction between TGF-β and the BG orphan domain

It was not previously known how the BG orphan domain engages the TGF-βs. Initial assumptions were drawn from the structure of the closest homolog, ENG, bound to BMP9 which binds with 2:1 stoichiometry and uses an edge β-strand to bind to BMP9 fingers (ref. 22). However, our structural data demonstrates a completely distinct interface, with

**Fig. 3 | The TGF-β2:BG$_{ZP-C}$ interface. a** Visualization of the binding interface between BG$_{ZP-C}$ and TGF-β2. Details of the interface are shown magnified below (panels I-III). The orientations are denoted by the sides of the cube. (I, II) Hydrophilic interactions between the extended regions of the fingers and the FG-loop of BG$_{ZP-C}$, and between the fingertips and CC'-loop and C'-strand are highlighted and denoted by dashed lines representing intermolecular bonds. (III) Hydrophobic contacts form between the concave surface of the fingers and the FG-helix that forms upon growth factor binding. Side chains of residues contributing to the hydrophobic interactions are highlighted in yellow. **b** Chart summarizing SPR data of BG$_{ZP-C}$ mutant variants. Mutations N82A, I161A, and A158E in BG$_{ZP-C}$ that completely abrogate binding to mmTGF-β2 are highlighted in red. Depicted mean K$_D$ values with standard deviation were derived from series of dilution in three consecutive repeats. **c** Selected SPR binding sensorgrams of BG$_{ZP-C}$ mutants binding to

immobilized TGF-β2 on the chip surface. Sensorgrams show a 2-fold dilution series of BG$_{ZP-C}$ variant injections (black). Fitting curves (red) are derived from data collected in three consecutive repeats. **d** AF2M model of TGF-β2 in complex with BG$_{ZP-C}$, with blowup panel highlighting potential interaction between BG$_{ZP-C}$ FG- and TGF-β2 heel helices. **e** Chart summarizing SPR binding data on selected combined mutants (Q57A, K60A, L64A of the heel helix and Y50A, L51A of the pre-helix extension) binding to BG$_{ZP-C}$ and zfBG$_O$. Depicted mean K$_D$ values with standard deviation were derived from series of dilution in three consecutive repeats. **f** The mutants that most strongly perturbed binding in (**b**) were further investigated using a CAGA$_{12}$-luciferase (CAGA-luc) assay for signaling activity in L6E9 cells. Statistical significance was determined using a two-way ANOVA test. Each point represents a separate biological replicate. ****$P$ < 0.0001. Source data are provided as a Source Data file.

the zfBG$_O$ subdomains, OD1 and OD2, and the paired antiparallel β strands that tether them together, forming an arc that straddles the edge of the TGF-β fingers (Figs. 4a and 1k). The positioning of zfBG$_O$ on TGF-β is responsible for the 1:1 stoichiometry as a significant clash between OD1 from both protomers would occur with a 2:1 stoichiometry (Fig. 4b). The most extensive and crucial interface for TGF-β binding is formed by the loop from residues 244–251 located between the β2 and β3 strands of OD1 with the sequence PNSSSAFQ (highlighted in orange in Fig. 4a I). This binding loop (BL) lies on the concave surface of the fingers and extends to the heel helix of the neighboring monomer (Fig. 4a I). The replacement of BL with an unrelated (scrambled, sc) sequence of similar length (scBL) resulted in complete loss of binding (Supplementary Fig. 6a). Residues N245, S247, and Q251 form hydrophilic interactions with fingers I and II of TGF-β, while A249 and F250 engage W30, W32, L101, M104 within the concave surface (Fig. 4a I) of the fingers and T57 in the proximal part of the heel helix of the opposing monomer (Fig. 4a III). Mutations in this region, such as F250A or D253A in zfBG$_O$ (Fig. 4c and Supplementary Fig. 6b), or in the heel helix of TGF-β that is contacted by the BL, such as the Q57A, K60A, L64A triple mutant of TGF-β2 described earlier (Fig. 3e and Supplementary Fig. 5a), either eliminated or significantly impaired binding. One caveat to these results is that the D253A mutation appeared to perturb the structure globally or increase protein aggregation, as revealed by 1D $^1$H NMR, though other substitutions in zfBG$_O$ did not (Supplementary Fig. 6c). On the other subdomain, OD2, we observed that residues L208, L210, and L213, which are located on the loop between β10 and the OD2 exit strand, pack against the outer convex surface of the fingers (Fig. 4a II). Mutation of any of these residues to alanine also dramatically reduced binding, with the latter two eliminating binding completely (Fig. 4c and Supplementary Fig. 6b). The interface is closed with F49 located at the end of the linker β-strand, mutation of which was also detrimental to ligand binding (Fig. 4c).

The identification of critical residues in the zfBG$_O$ interface paved the way for a deeper understanding of the specificity of zfBG$_O$ towards TGF-β family ligands. Inspection of the interface buried area and TGF-β family Multiple Sequence Alignment (Fig. 4d), led to recognition of potential residues characteristic to only TGF-β isoforms, namely E99, L101, S102, or N103. Of note, the Q57A mutation in the heel helix of the adjacent monomer was probably the key residue in the triple mutant that impacted BG$_O$ binding (Fig. 3e). However, this residue is not conserved among TGF-β isoforms, suggesting that the proper positioning of the heel helix is essential. The AF2M simulations indicated that while substitution of each of these residues with the equivalent residue in BMP2 only moderately diminished zfBG$_O$ binding, combinations of these amplified the disruptive effect, with the quadruple variant of E99, L101, S102, and N103 placing the complex prediction score among the non-binders (Fig. 4e). Unfortunately, it was not possible to evaluate the effect of these substitutions in vitro, as this quadruple variant and related triple or double variants could not be isolated, suggesting that these residues must be playing further structural roles.

## The mechanism behind signal potentiation
Co-receptors, while not capable of signaling on their own, enhance the signaling potency of ligands via their primary receptors, which has important implications for the precise modulation of cellular signaling[3,4,15]. There are several mechanisms through which co-receptors can act, including increasing the effective ligand concentration at the plasma membrane[22,48,49], potentiation of signaling receptor binding by allosteric modulation[50] or by direct contact with the signaling receptor[51], or mediation of receptor internalization[52].

Signal potentiation by ENG is thought to rely on efficient capture of BMP9/10 on the cell surface, increasing the local concentration and propensity of the ligands to bind and recruit their type I receptor, ACVRL1 (ref. 22). It was suggested that BG not only potentiates signaling using the same mechanism as ENG, but also by indirectly or directly increasing the binding affinity between TGF-βs and TGFBR2 through its orphan domain[7]. Our structure of BG bound to TGF-β supports this dual mechanism as BG embraces the ligand homodimer in a way that allows one molecule of TGFBR2 to bind, thus allowing for the signaling receptor recruitment to the complex by simply "presenting" the ligand to the signaling receptor. In addition, the proximity of bound BG$_O$ to recruited TGFBR2 suggests possible contact between the co-receptor and receptor (Fig. 1h) and thus direct potentiation. To confirm the direct potentiation mechanism, we conducted an SPR potentiation assay using both zfBG$_O$ and ratBG$_O$. It was surprising to discover that in contrast to ratBG$_O$, zfBG$_O$, which we used for X-ray crystallography, did not potentiate the binding of TGF-β2 to TGFBR2 (Fig. 5a–c). ratBG$_O$ and zfBG$_O$ share 55.4% sequence identity, with no obvious difference that we could identify to explain the observed effect. To decipher the direct potentiation mechanism, we analyzed the TGF-β1:BG$_O$:(TGFBR2)$_2$ complex by Cryo-EM where zfBG$_O$ was replaced with ratBG$_O$ (Supplementary Fig. 7a–c). The strong preferential orientation observed previously was overcome using graphene oxide grids (Supplementary Fig. 7d) enabling building of a model into the 3.40 Å resolution Cryo-EM map (Supplementary Table 2). From the structure it was apparent that the most pronounced difference lay within the TGF-β binding loop of OD1, where there is a two-residue insertion in ratBG$_O$ compared to zfBG$_O$ (Fig. 5d, e) that allows the BL in ratBG$_O$ to make additional contacts with TGFBR2. To test our model, we swapped the shortened loop from zfBG$_O$ into ratBG$_O$. Despite similar binding affinity for TGF-β2 (Supplementary Fig. 7e), potentiation of TGFBR2 binding was not detected (Fig. 5f, h), showing that the extension in the binding loop is essential for potentiation. In the structure of the TGF-β2:ratBG$_O$:(TGFBR2)$_2$ complex, Y248 is predicted to be responsible for most of the contact with TGFBR2 residues F10, P11, L13 and I39 (Fig. 5e). Indeed, mutation of ratBG$_O$ Y248A nearly eliminated the potentiation effect, in accordance with our model (Fig. 5g, h). This unexpected difference between zfBG$_O$ and ratBG$_O$ highlighted the importance of the BL of ratBG$_O$ not only for efficient binding to TGF-βs, and thus indirect potentiation of binding of the signaling receptors by increasing the local concentration, but also for direct potentiation of TGFBR2 binding. This dual mechanism

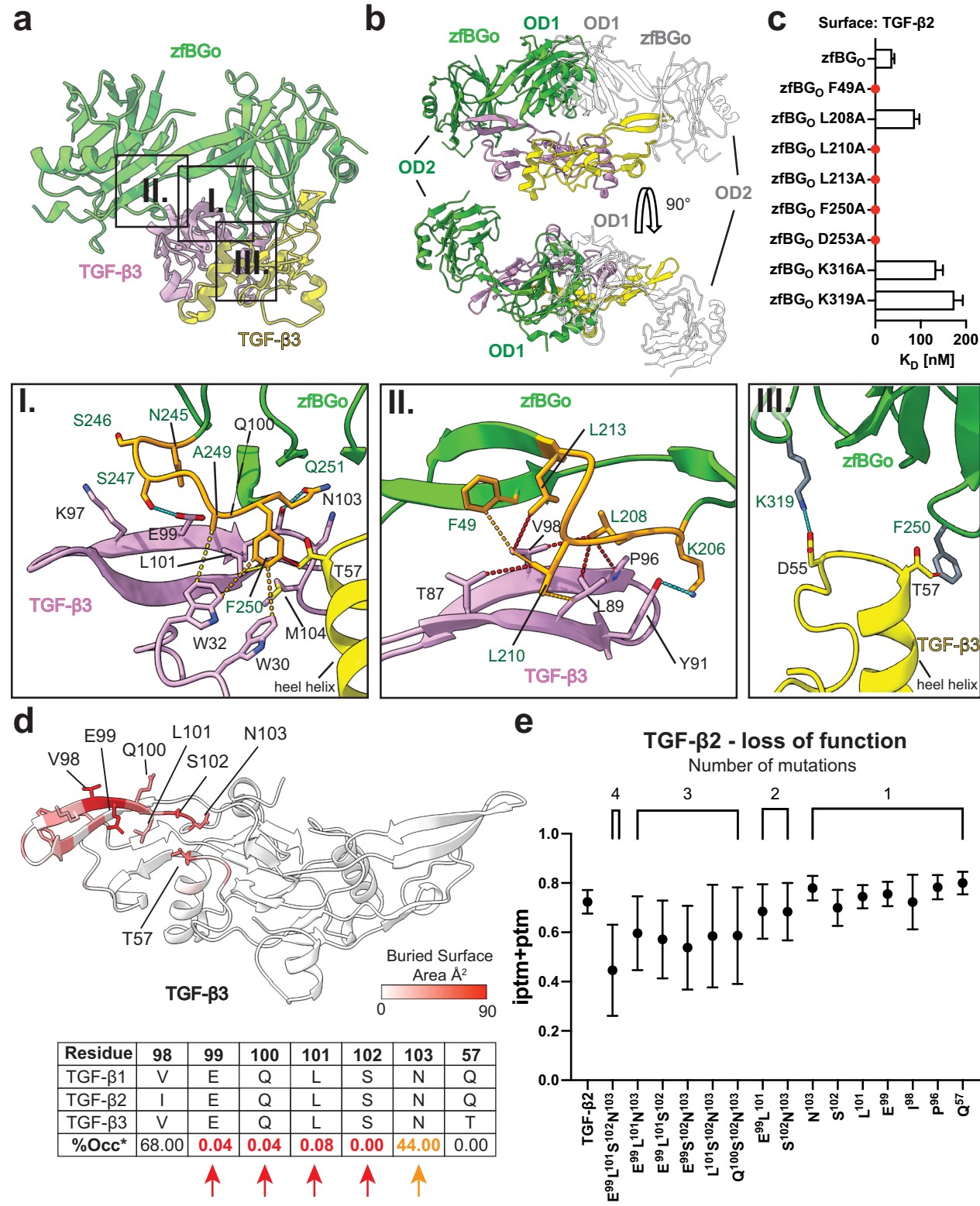

allows for more equal binding of all TGF-β isoforms to the signaling receptors.

## InhA competes with TGF-β through binding to distinct but overlapping regions of BG_{ZP–C}

InhA and InhB play crucial roles in endocrine signaling pathways that regulate fertility[18]. InhA and InhB are heterodimers which share a common α-subunit but differ in their β-subunit composition. InhA

contains a βA subunit encoded by the *INHBA* gene, which is also shared with ActA, while InhB contains a βB subunit encoded by *INHBB*, which is shared with activin B (ActB). InhA binds the activin type II receptors, ACVR2A or ACVR2B, through its βA subunit. In cells, this interaction is further promoted by the binding of BG_{ZP-C} to the α-subunit, effectively sequestering ACVR2A or ACVR2B from forming a signaling complex with ActA and the activin type I receptor, ACVR1B. Despite sharing the same α subunit with InhA, InhB has

**Fig. 4 | The TGF-β:zfBG$_O$ interface. a** The TGF-β:zfBGo interface (PDB: 9B9F, regions highlighted in panels below) is mostly maintained through hydrophobic interactions (I, II) between the ligand fingers and the binding loop (BL) (highlighted in orange) present on OD1 (I) and a second, shorter adjacent loop present on OD2 (highlighted in orange; II). (III) Interactions between zfBG$_O$ and the heel helix (indicated) of the second monomer of TGF-β1. Hydrophobic interactions between zfBG$_O$ and TGF-β3 are depicted as pseudobonds in red (< 4 Å), orange (4–4.5 Å) or yellow (4.5–4.8 Å). Selected hydrogen bonds are depicted in blue (< 4 Å). **b** Conformation of zfBGo bound to TGF-β1 explains the 1:1 stoichiometry as the second protomer of zfBG$_O$ (white) cannot occupy binding area on the second monomer of TGF-β1 (yellow) due to significant structural clash between OD1 subdomains of both protomers. **c** Chart summarizing SPR binding data on selected interface mutants of zfBG$_O$. Mutations F49A, L210A, L213A, F250A and D253A in zfBGo that completely abrogate binding to mmTGF-β2 are highlighted in red. Depicted mean K$_D$ values with standard deviation were derived from series of dilutions in three repeats. **d** Structure of TGF-β3 colored after Buried Surface Area (BSA) demonstrating contribution of each residue to the binding interface when in complex with zfBG$_O$ (PDB: 9B9F). The calculation was performed using the PDBe-PISA webserver. Below, a table showing residue conservation among TGF-β iso-forms and % of the occurrence (%Occ) of that residue in other growth factors from the TGF-β family. **e** Selected residues in TGF-β2 based on PISA calculations were mutated *in-silico* into the equivalent BMP2 residues to assess their contribution to binding by AF2M. Single, double, triple and quadruple combinations were tested, showing their additive effect and the most disruptive combination. The y-axis represents the iptm+ptm score mean values with standard deviation derived from a total of 25 models generated for each complex. Source data are provided as a Source Data file.

only a modest binding affinity for the BG$_{ZP-C}$ domain[27]. Recent studies have shown that InhB primarily exerts its effects in gonadotrope cells through a TGFBR3-like protein, TGFBR3L, diverging from the pathway utilized by InhA[27].

These observations suggest that both the α and β subunits of InhA may mediate binding to BG$_{ZP-C}$. We therefore attempted to fit InhA into the TGF-β binding site such that the α subunit of InhA aligns similarly to one monomer of TGF-β2. In this arrangement, the β subunit would be able to contribute to the binding process in a manner comparable to how the second monomer of TGF-β2 interacts, specifically through the heel helix and FG-helix of BG$_{ZP-C}$. To test our hypothesis, we employed NMR spectroscopy, focusing on the BG$_{ZP-C}$ domain with ¹³C-methyl-Ile labelling. This technique allows detection of distinct 'fingerprints' of the BG$_{ZP-C}$ domain both unbound and bound to the ligand, as depicted in Fig. 6a, b. Our data show that both the TGF-β2 dimer and mmTGF-β2 generate highly similar patterns of chemical shift changes, including the transition of I161 from a random coil to a non-random coil chemical shift, consistent with the disordered loop to helix transition observed in the X-ray structure. Upon titration of ¹³C-methyl-Ile BG$_{ZP-C}$ with InhA, we observed a concentration-dependent weakening of signal intensities and peak broadening, which suggests binding (Fig. 6c–e). However, peak broadening may also occur from oligomerization or aggregation, making it challenging to draw definitive conclusions about the position of the binding interface. Therefore, we conducted a cell-based competition experiment in which InhA and TGF-β directly compete for binding to the BG$_{ZP-C}$ domain (Fig. 6f, g). In this experiment, we labeled InhA with a fluorescent marker and observed clear co-localization with SNAP-BG, which was stably expressed in our HEK293T cell line (Fig. 6f, top). This co-localization was disrupted by the addition of an excess of unlabeled InhA or TGF-β1, indicating that InhA forms a complex with SNAP-BG and that InhA and TGF-β compete for binding to BG$_{ZP-C}$ (Fig. 6f, rows 2 and 3). mmTGF-β2 also disrupted the co-localization, though consistent with its reduced binding affinity for BG$_{ZP-C}$, it competed with reduced potency (Fig. 6f, row 4) and required a higher ratio to obtain similar results (Supplementary Fig. 8a). ActA did not diminish the co-localization of InhA with BG$_{ZP-C}$, consistent with its inability to bind BG$_{ZP-C}$ (Fig. 6f, row 5).

In addition, we also investigated how mutation of specific residues on BG$_{ZP-C}$ affect binding to InhA relative to TGF-β (Fig. 6h). Interestingly, mutations such as N82 or I161 to alanine, and even more severe alterations, such as N82 or I161 to tyrosine, that disrupted binding to TGF-β2 did not significantly affect binding to InhA. The only mutation that caused a modest reduction in binding was A158 to glutamine (Fig. 6h, i and Supplementary Fig. 8b, c). Collectively, these results indicate that InhA binds to BG$_{ZP-C}$ through an interface that partially overlaps with the one determined for TGF-β. However, the exact position or involvement of structural features on BG$_{ZP-C}$ that enable binding likely differ.

## BG does not release TGF-β2 from the latent complex in solution

Considering the mechanism of ligand–co-receptor assembly, we explored the potential role of BG in facilitating the release of mature TGF-β2 from its latent complex. Although we initially hypothesized that BG might aid in this release, the structures of TGF-β in complex with both BG$_{ZP-C}$ and BG$_O$ suggested this was unlikely. The mature TGF-β ligands bind the pro-domain[53–55] using similar residues that are involved in binding to BG, suggesting that the pro-complex shields the mature growth factor from BG via the lasso loops and straitjacket domain[53]. Consistent with this, we were not able to detect a complex between pro-TGF-β2 and either BG$_O$ or full-length BG (Supplementary Fig. 9a, b). Furthermore, we did not observe sequestration or displacement of the TGF-β2 from the latent complex by either BG$_O$ or full-length BG in vitro (Supplementary Fig. 9c, d).

## Discussion

Here, we present the structure of full-length BG bound to TGF-β, offering insights into BG's specificity of ligand binding and mechanism of signal potentiation. We combined X-ray crystallography, Cryo-EM, and NMR spectroscopy which allowed us to study the binding interfaces in fine detail, in the context of the full-length co-receptor and in the presence of the signaling receptors. The determined structures show how the BG$_O$ and BG$_{ZP-C}$ domains bind independently to TGF-β dimers, allowing for efficient recruitment of TGF-β to the cell surface and how the signaling receptors take over the ligand and thus signal (Fig. 7a). Using our cell-based fluorescent assay and advanced modeling with AF2M, we also showed the remarkable BG specificity towards specific members of TGF-β ligand family (Fig. 7b) and using both site-directed mutagenesis and modeling we demonstrate how this specificity is achieved. Lastly, leveraging structural, biophysical, and cell-based assays we also provide insights into how BG engages InhA, which is distinct from the TGF-β isoforms, but nevertheless competes with them for binding (Fig. 7c).

### The mechanism by which BG potentiates TGF-β signaling

The high-resolution structures of the zfBG$_O$ and BG$_{ZP-C}$ domains bound to TGF-β, together with the low-resolution structure of the TGF-β1:BG complex, show that BG$_O$ binds TGF-β such that the adjacent TGFBR2 binding site is still accessible for binding. In contrast, the BG$_{ZP-C}$ domain blocks the binding sites of both TGFBR1 and TGFBR2. Thus, once BG binds and captures TGF-β ligands on the cell surface, this likely initiates the recruitment of one TGFBR2 at the one receptor binding site that remains accessible (Fig. 7a). The structure of the TGF-β1:BG complex further shows that BG$_{ZP-N}$ neither participates in binding nor is rigidly tethered to BG$_{ZP-C}$. This explains indirect observations that the BG$_{ZP-N}$ domain is not required for TGF-β potentiation[4].

The structural and biophysical binding data demonstrate that the binding loop (BL) of ratBG$_O$ is not only the main structural element required for binding to the TGF-βs but also potentiates the binding of TGFBR2 through direct co-receptor–receptor contact. The

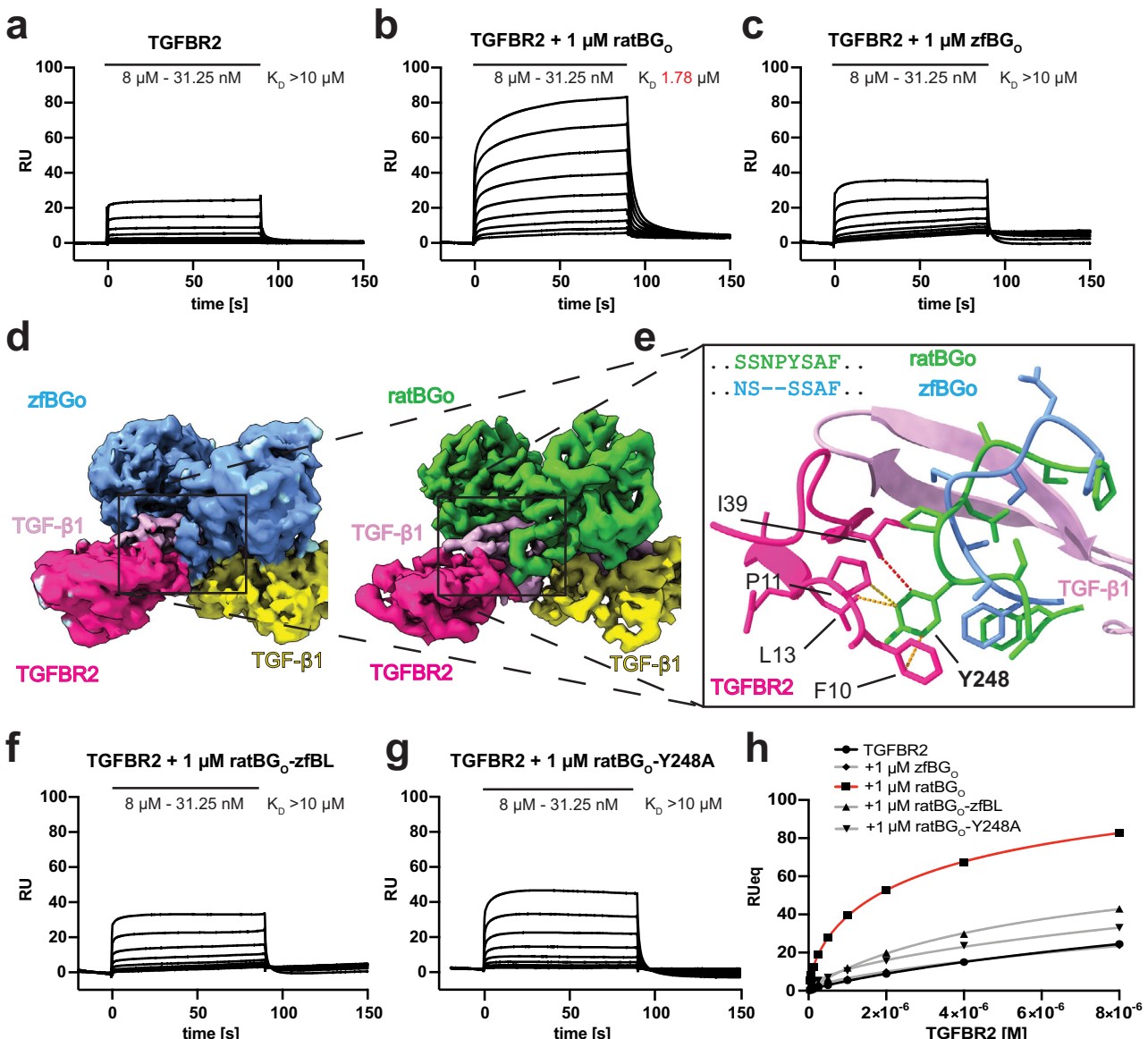

**Fig. 5 | ratBGo directly contacts and potentiates binding of TGFBR2. a–c** SPR sensorgrams showing response to a 2-fold dilution series of injections of TGFBR2 over immobilized TGF-β2. The same experiment was repeated in the presence of 1 μM ratBG$_O$ (**b**) or 1 μM zfBG$_O$ (**c**). All experiments were repeated at least three times. **d, e** Zoom into TGF-β binding loop regions of Cryo-EM structures of zfBG$_O$ (PDB: 9FKP) and ratBG$_O$ (PDB: 9FDY) in complex with TGF-β1 dimer and two TGFBR2 ECDs (**d**). A blow-up of the boxed area demonstrates how ratBGo, but not zfBGo, can form additional contact with TGFBR2 through Y248 (**e**). Selected hydrophobic interactions between zfBG$_O$ and TGFBR2 are depicted as pseudo-bonds in red (< 4 Å), orange (4–4.5 Å) or yellow (4.5–4.8 Å). SPR sensorgrams showing response to a 2-fold dilution series of injections of TGFBR2 over immobilized TGF-β2 in the presence of 1 μM ratBG$_O$-zfBL (**f**) and 1 μM ratBG$_O$-Y248A (**g**). All experiments were repeated at least three times. **h** Plot of the normalized equilibrium responses of TGFBR2 binding to TGF-β2 in the presence of various ratBG$_O$ variants. Presented curves are fitted to datapoints collected from three independent repeats. Source data are provided as a Source Data file.

potentiation is mediated by expanding the binding interface and increasing the affinity, not through allostery. The two amino acid deletion occurring in zfBG completely impairs the direct, yet not indirect, potentiation of TGFBR2 binding and is also present in most other boney fish families. In contrast, the extended version is present not only in amphibians, reptiles, birds, and mammals, but also in cartilaginous fish and primitive boney fish from the *Polyodontidae*, *Acipenseridae*, and *Polypteridae* families. This pattern suggests a divergent evolution of the BG protein in teleosts, rendering it a possibly less transferable model for studying TGF-β signaling by BG in other vertebrates, as previously noted[56].

The transition from the TGF-β:BG:TGFBR2 ternary complex to the signaling complex has been proposed to occur by a handoff mechanism, whereby the one bound TGFBR2 potentiates the recruitment of TGFBR1 through direct receptor–receptor contact, and in the process displaces the orphan domain, leaving the second monomer of TGF-β bound by only the BG$_{ZP-C}$ domain (Fig. 7a)[4,7]. Inspection of the TGF-β:BG$_O$:(TGFBR2)$_2$ and the TGF-β3WD:zfBG$_O$:TGFBR2:TGFBR1 structures shows that the BL of BG$_O$ partially occupies the TGFBR1 interface formed by the heel helix of the adjacent TGF-β monomer and TGFBR2, which would be expected to block binding of TGFBR1. However, the binding affinity of TGFBR1 to the TGF-β:TGFBR2 binary complex (K$_D$ ca. 30 nM)[41] is significantly higher than BG$_O$ binding to TGF-β (K$_D$ ca. 200 nM)[7], enabling TGFBR1 to outcompete and displace BG$_O$. It is notable that the heel helix of the second monomer of TGF-β also takes part in binding of BG$_O$, albeit through a limited interface formed mostly by two residues, D55, T57 (Fig. 4a III, d), located on the solvent-exposed surface of the heel helix.

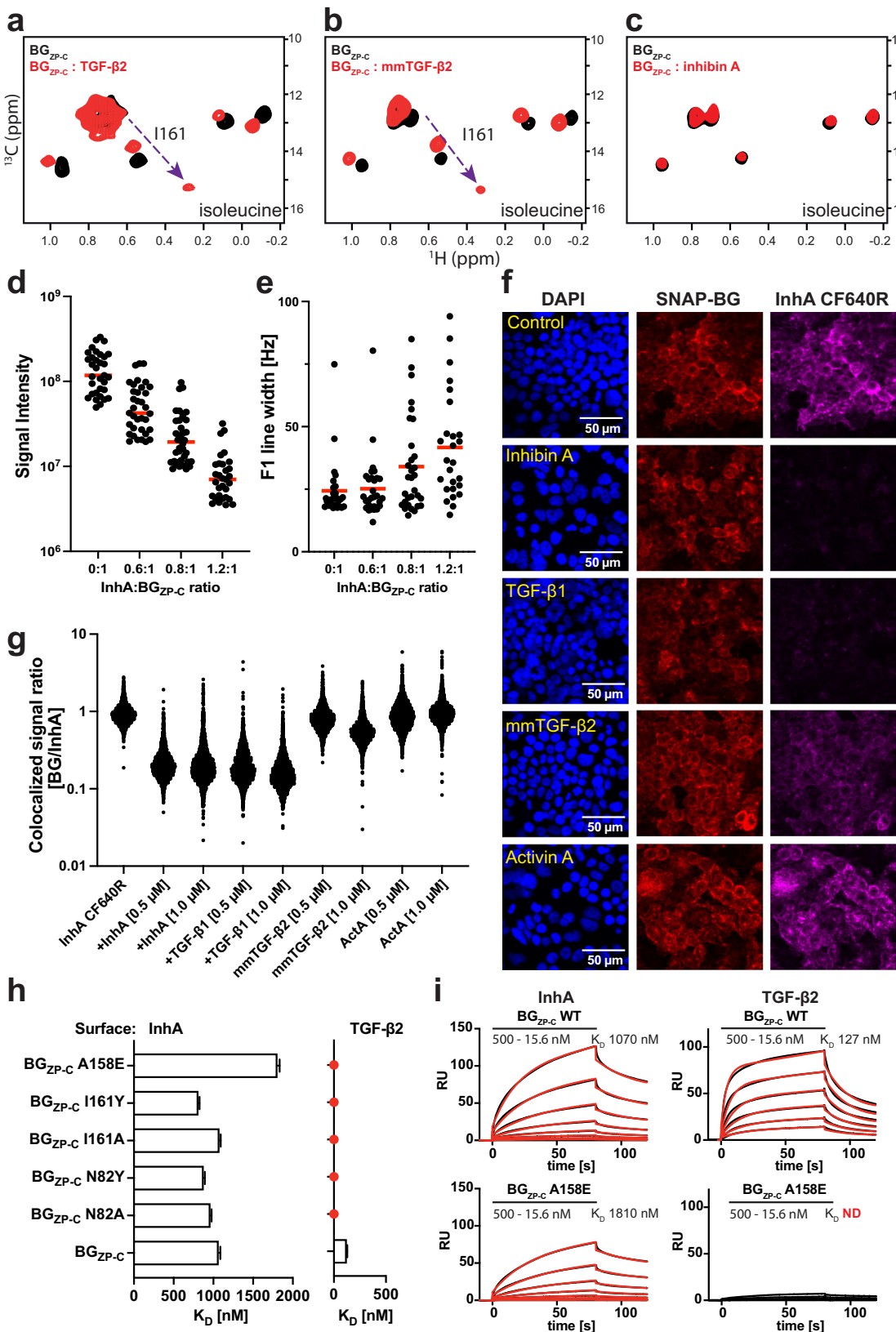

Mutations in this region of TGF-β2 almost completely abrogate $BG_O$ binding (Fig. 3e and Supplementary Fig. 5a) and are likely responsible for the diminished binding of mmTGF-β2 (ref. 25). This suggests that relatively small changes close to the heel helix may have a pronounced effect on $BG_O$ binding and thus partial docking of TGFBR1 may facilitate the rapid release of $BG_O$ that drives the handoff (Fig. 7a IV-V).

However, it remains unclear how TGFBR2 or TGFBR1 can outcompete $BG_{ZP-C}$ from its binding position. It is proposed that the complete displacement of BG would be beneficial for signaling, as it has been demonstrated that a heterodimer of TGF-β3, capable of forming a signaling complex on only one side, can still initiate signaling but the amplitude of the signal measured by SMAD3 phosphorylation

**Fig. 6 | InhA and TGF-β interact with partially overlapping, but distinct regions of BG_{ZP-C}. a–c** Isoleucine δ1 methyl region of the $^1$H–$^{13}$C correlation NMR spectra of BG$_{ZP-C}$ (black). Upon TGF-β2 (**a**) or mmTGF-β2 (**b**) binding (red), the chemical shift of I161 δ1 methyl is perturbed, with the signal shifting from the random coil to the structured region, consistent with a disordered loop to helix transition. No shifts were observed upon binding of InhA (**c**). Representative measurement of signal intensity (**d**) and line width (**e**) changes of methyl signals upon InhA titration into the Ile δ1, Met, Val and Leu methyl-labeled BG$_{ZP-C}$ sample. Each dot represents a line width of a single residue upon titration. Mean value of signal intensity or line width within a sample is represented by a red bar. The measurement was repeated twice. **f** Confocal microscopy images of competition assay - HEK293T cells stably transfected with a SNAP-tagged BG construct (SNAP-BG) stained with DAPI (blue, left panels), incubated with a SNAP-tag ligand conjugated to the CF567 fluorescent dye (red, middle panels), along with InhA tagged with CF640R fluorescent dye (violet,

right panels) alone (row 1) or with the addition of the indicated unlabeled ligands (rows 2–5). Bound InhA could be outcompeted by addition of excess unlabeled InhA, TGF-β2 or mmTGF-β2. Scale bar, 50 μm. **g** Quantification of the fluorescent colocalized SNAP-BG/InhA signals. Each data point represents the ratio of SNAP-BG signal to InhA signal within a segmented region of the image generated by applying the Otsu thresholding method to the SNAP-BG channel. Analysis was performed on three images per condition, each corresponding to an independent biological replicate ($n = 3$). **h** Chart summarizing SPR binding experiments using selected panels of BG$_{ZP-C}$ mutants binding to immobilized TGF-β2 or InhA. Depicted mean K$_D$ values with standard deviation were derived from series of dilution in three consecutive repeats. **i** Selected SPR binding sensorgrams show response to a 2-fold dilution series of injections of BG$_{ZP-C}$ variants (black) over TGF-β2 or InhA immobilized on the chip surface. Fitting curves (red) were generated based on three consecutive repeats. Source data are provided as a Source Data file.

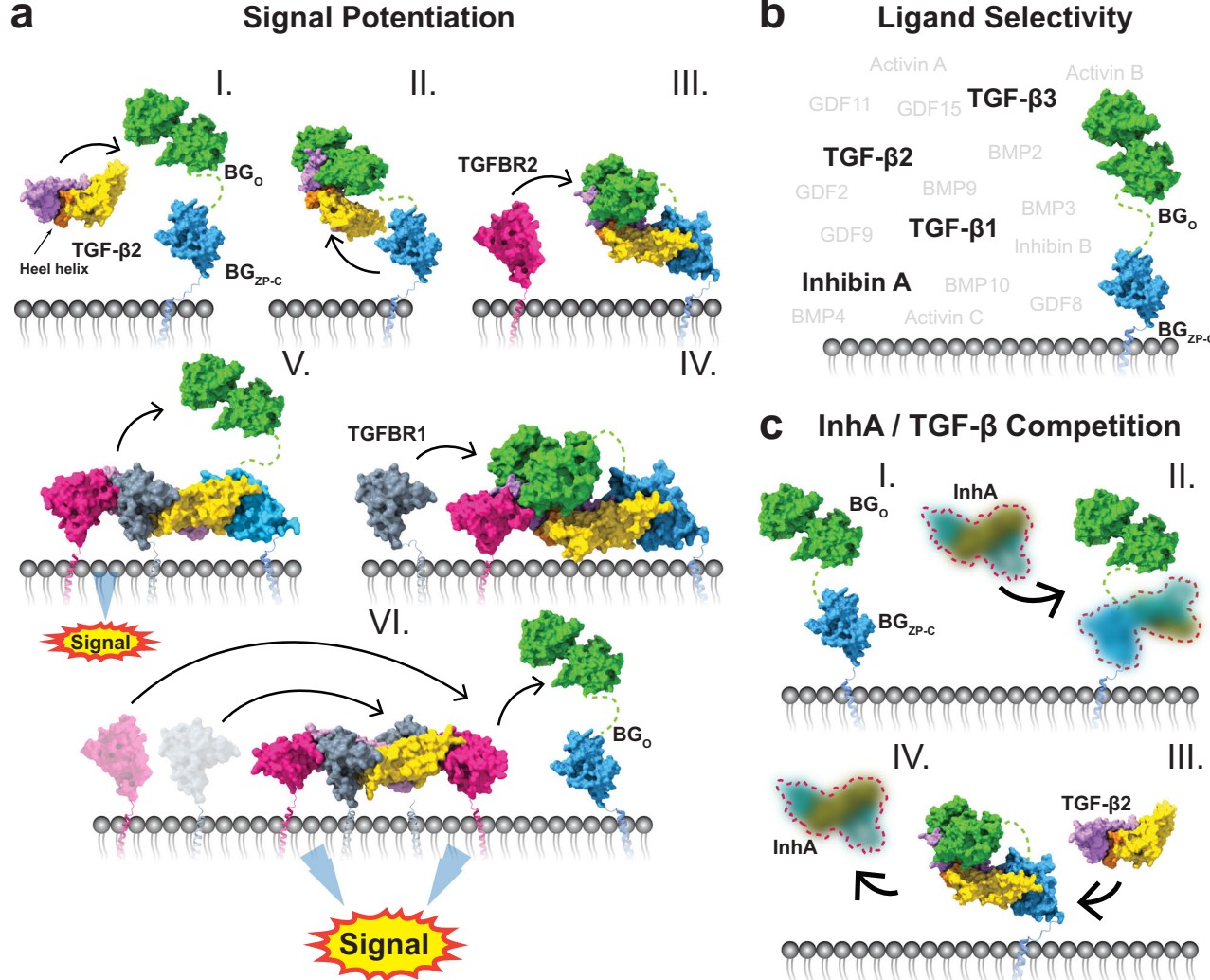

**Fig. 7 | Structural illustration of BG ligand selectivity and TGF-β2 signaling enhancement. a** Mechanism of TGF-β2 signal potentiation by BG. In the first step TGF-β2 binds to BG$_O$ (I), then the BG$_{ZP-C}$ domain binds (II). The presence of BG$_O$ increases the affinity towards TGFBR2 (III) that binds to the available site adjacent to the BG$_O$ domain. Next, TGFBR1 is recruited by binding to the shared TGF-β2:TGFBR2 interface (IV), which displaces BG$_O$ (V). This complex can already signal.

However, BG$_{ZP-C}$ can be further displaced and the heterotetrameric signaling complex, with full signaling potential, can be formed (VI). **b** Schematic showing BG ligand selectivity towards TGF-β isoforms as well as InhA among other TGF-β family members. **c** InhA competes with TGF-β2 for binding to BG$_{ZP-C}$. BG$_{ZP-C}$ binds to InhA (blurred picture denotes lack of determined molecular structure) and forms a complex (I, II). TGF-βs can compete off InhA from its binding position (III, IV).

is significantly reduced compared to the wild-type homodimer[41]. The structure of the mmTGF-β2:BG$_{ZP-C}$ complex shows that BG$_{ZP-C}$ not only binds and engages the two critical basic residues on the tips of fingertips where TGFBR2 binds, but also occupies the underside of the fingers where TGFBR1 binds. In modeling studies it was shown that in

spite of the lower affinity of TGFBR2 and TGFBR1 for binding TGF-β2 compared to BG$_{ZP-C}$, together these can nonetheless displace BG$_{ZP-C}$ to form the TGF-β2:(TGFBR2)$_2$:(TGFBR1)$_2$ signaling complex, due to transient disassociation of BG$_{ZP-C}$ and sequential binding of TGFBR2 then TGFBR1 or TGFBR1 then TGFBR2 (ref. 37). There are two

mechanisms that could further promote this manner of assembly. The first is internalization of the TGF-β2:BG:TGFBR1:TGFBR2 complex into endosomes, which may occur based on a previous report that the disordered but highly conserved cytoplasmic tail of BG is bound by β-arrestin which facilities internalization[57]. Then, assembly might follow a mechanism demonstrated for the repulsive guidance mediators (RGMs), co-receptors for some BMPs, whereby internalization in endosomes triggers the disassociation of the co-receptor due to a lowering of the pH, followed by binding of the signaling receptors[52,58]. The second possible mechanism is that proteolytic cleavage might occur within the $BG_{ZP-C}$ FG-loop/helix[9] thereby weakening binding and enabling recruitment of a second pair of signaling receptors (Fig. 7a VI).

It is also possible that the BG is not fully displaced and the ligand signals as a quaternary complex with $BG_{ZP-C}$ still bound. This is suggested by the finding that TGF-βs bound to a single TGFBR2:TGFBR1 heterodimer can signal as described above. However, in the aforementioned modeling studies[37], this manner of binding could not recapitulate the biphasic effect on TGF-β2 signaling, whereby low levels of BG potentiate signaling, while at higher levels, BG has the opposite effect. Even though the biphasic behavior is well supported by experimental data, there is no direct experimental evidence showing a transition point where BG would turn from potentiation to inhibition as a function of BG levels. Therefore, the second phase where BG diminishes TGF-β2 signaling could be attributed to domain shedding[59]. This explanation is further suggested by the detection of BG:TGF-β2:TGFBR2 complexes in crosslinking experiments[4], suggesting the non-transient nature of this complex.

### Role of BG in TGF-β2 release from latent complexes

In vitro experiments showed that in solution BG does not facilitate the release of mature TGF-β2 from its latent complex. It is not surprising as the structural analyses of TGF-β bound to $BG_{ZP-C}$ and $BG_O$ indicate that mature TGF-β ligands interact with their pro-domain through residues that also mediate BG binding, suggesting that the latent complex shields TGF-β from BG. However, it is worth noting that, in a cellular context where additional factors such as shearing forces may influence interactions, the release or partial release of mature ligand from latent depots remain a possibility[55,60]. Hence, while we could not demonstrate this interaction in vitro, we acknowledge that in a cellular context the dynamics of the ligand–co-receptor assembly may differ.

### BG ligand binding specificity

BG demonstrated remarkable selectivity towards the three TGF-β isoforms and InhA as assessed by our competition assays. Modeling with AF2M allowed us to broaden our investigation to include interactions between BG and all ligands of the TGF-β family. The experimental data and predictions from AF2M identified only the three TGF-β isoforms as binding partners of $BG_O$. Inspection of the binding interface, together with the multiple sequence alignment, highlighted residues likely responsible for ligand selectivity. However, we were not able to evaluate their significance using mutagenesis, as substitution of these amino acids disrupted the protein structure. Despite this, their importance was confirmed by AF2M, indicating that these residues are likely important determinants of the specificity.

For the $BG_{ZP-C}$ domain, the key residues suggested by the mmTGF-β:$BG_{ZP-C}$ structure, such as N82 and I161, were shown to be essential for binding, though most of the hydrophilic interactions occur between the side chain of one of the partners and the backbone of the other. The backbone atoms of closely related isoforms are expected to have a more consistent arrangement than the sidechains, which may enable the co-receptor to form stable interactions with more than one partner that share the same geometry but differ in sequence. The previously reported binding site residues that differ from those demonstrated by our structure[29,33] likely arose due to insufficiently validated mutants, as

some of the mutations likely impacted folding and/or trafficking to the membrane, rather than directly impacting ligand binding.

$BG_{ZP-C}$ is known to bind all TGF-β isoforms, but also InhA[17,25,27]. InhA binds to $BG_{ZP-C}$ through an interface that at least partially overlaps with the TGF-β interface, yet binding of InhA to $BG_{ZP-C}$ is insensitive to most of the mutations that abrogate binding to TGF-β. This suggests that $BG_{ZP-C}$ binds InhA differently to TGF-β, though the precise manner of binding is unclear, as models of the InhA:$BG_{ZP-C}$ complex generated using AF2M were variable and could not simultaneously account for both the competition and mutagenesis data. It is notable that for AF2M predictions of $BG_{ZP-C}$ complexes, the score gap between binders and non-binders is significantly narrower compared to the predictions for $BG_O$, lowering the confidence of the predictions. Moreover, no structure of InhA is available, thus modeling is more challenging, and obtaining a structure of the InhA:$BG_{ZP-C}$ complex will be essential. Our work does not rule out the possibility that BG can engage some other ligands of the TGF-β family, as suggested by some published data[44,45,61]. However, if such complexes form, these are likely significantly weaker than the complexes formed with the TGF-β isoforms or involve interfaces different from those described here.

### Structural insights into TGF-β pathway inhibition

The TGF-β pathway has been pharmacologically targeted using several strategies, including small molecule receptor kinase inhibitors, peptides, antisense oligonucleotides, neutralizing antibodies and ligand traps based on fusing of the extracellular domains of the TGF-β receptors. It is worth noting that TGFBR2-based traps, such as AVID200 (ref. [62]) or Bintrafusp Alfa[63] or neutralizing antibodies, such as GC1008 or GC2008 (ref. [64]), would be expected to compete for binding with membrane-bound BG and demonstrate attenuated efficacy in environments highly enriched with BG. Indeed, during our structural studies we attempted to utilize the neutralizing antibody GC2008 to increase orientational sampling for Cryo-EM. However, it was not possible to obtain a complex with both BG and GC2008 bound to TGF-β, as the components competed with one another.

One interesting strategy for inhibiting TGF-β is the use of a short, receptor-derived peptide that would block the interaction of the ligand with the receptor. One such example is the 14-mer peptide P144, whose sequence is derived from the $BG_{ZP-C}$ domain of human BG. It was shown that P144 blocks TGF-β1 biological activity in different in vitro and in vivo models[34,65–67] and it currently is the drug candidate Disitertide®, which is being tested for pathological skin fibrosis, such as scleroderma. Indeed, our mmTGF-β2:$BG_{ZP-C}$ structure demonstrates that the sequence is precisely derived from the loop region that forms the FG-helix upon binding to concave surface of the fingers of the growth factor[65]. We anticipate that our structural data will advance the knowledge-based design of next-generation inhibitors, enhancing both their potency and biochemical properties.

## Methods

### Materials

Details of all materials (antibodies, bacterial strains, cell lines, plasmids, recombinant proteins, chemicals etc) used in this study as well as software is given in Supplementary Table 3.

### Cloning

All constructs used to express proteins for X-ray Crystallography and Cryo-EM were already available from previous studies (Supplementary Table 3). For SPR binding studies a set of mutants was prepared using site-directed mutagenesis of pCDNA3.1 + -ALB-$BG_{ZP-C}$-(N82A, N82Y, K86A, D88A, E130A, L143A, S155A, A158A, A158E, I161A, I161Y, W162A, M164A, M165, W162A/M165A or Q166A); pCDNA3.1 + -zfBG$_O$-(F49A, L208A, L210A, L213A, F250A, D253A, K316A or K319A) or by cloning respective GenBlocks into pCDNA3.1+ vector (IDT DNA) to generate

pCDNA3.1-ratBG$_O$-zfBL, pCDNA3.1-ratBG$_O$-Y248A, pCDNA3.1-ratBG$_O$-BL, pCDNA3.1+-proTGF-β2, pCDNA3.1+-proTGF-β2-(E99K, L101Y, S102Q, N103D), pCDNA3.1+-proTGF-β2-(E99K, S102Q, N103D), pCDNA3.1+-proTGF-β2-(E99K, S102Q), pCDNA3.1+-proTGF-β2-(S102Q, N103D), or into the pET32a vector to generate pET32a-TGF-β2-(Q57A, K60A, L64A), pET32a-TGF-β2-(Y50A, L51A) using In-Fusion seamless cloning (Takara Bio). SNAP-BG was constructed using overlapping PCR, joining the SNAP domain with a C-terminal linker "GGGGSGGGGSGGG" to the N-terminus of rat BG (residues 30–853) and cloning the entire construct into the pCDNA3.1+ vector. The CAGA$_{12}$-luciferase reporter[47] and the BG$_{ZP-C}$ construct[19] used in luciferase assays were described previously. The mutant BG$_{ZP-C}$ constructs were constructed using the QuikChange protocol with the following primers:

N82A_Forward-TGATTACACCATCATCGAGGCCATCTGTCCGAAAGACGAC, N82A_Reverse-GTCGTCTTTCGGACAGATGGCCTCGATGATGGTGTAATCA, I161A_Forward-CTCTCGATGCCACCATGGCCTGGACCATGATGCAGA, I161A_Reverse-TCTGCATCATGGTCCAGGCCATGGTGGCATCGAGAG.

## Expression and purification of TGF-β growth factors

The growth factor domains of human TGF-β2, TGF-β3, TGF-β3WD and mmTGF-β2 were produced as described previously[38]. Briefly, proteins were expressed in *E. coli* BL21(DE3) in the form of insoluble inclusion bodies and after washing, the material was reconstituted in 8 M urea, reduced using DTT and refolded by dilution into non-denaturing buffer. The natively folded growth factors were isolated by high-resolution cation-exchange chromatography using a MonoS column. Intact masses of the purified proteins were measured by LC electrospray ionization TOF MS (LC-ESI-TOF-MS, Bruker Micro TOF, Billerica, MA) and folding was assessed by 2D NMR. Pro-TGF-β1 was produced in expi293F cells with expi293™ expression medium and mature growth factor was purified from the pro-domain by high-resolution cation-exchange chromatography on a MonoS column (GE Healthcare).

## Expression and purification of mammalian-expressed BG$_{ZP-C}$ and BG$_O$

Rat BG$_{ZP-C}$ and the rat or zebrafish orphan domains, (ratBG$_O$, zfBG$_O$, respectively) were expressed as secreted proteins in suspension-cultured expi293F cells with expi293™ expression medium, as described previously[7]. To gain a better yield, the BG$_{ZP-C}$ domain was expressed fused to albumin, which was removed by proteolytic cleavage with thrombin and separated by size exclusion chromatography.

## Expression and purification of bacterially expressed BG$_{ZP-C}$ domain

An isotopically labeled sample of BG$_{ZP-C}$ for NMR studies was expressed in *E. coli* BL21(DE3) cells (EMD-Millipore) using the plasmid pET32a (EMD-Millipore), where it was positioned downstream of thioredoxin, a hexahistidine tag, and a thrombin cleavage site. The coding sequence for BG$_{ZP-C}$, corresponding to residues 590–757 of NCBI NP_058952, included a StrepII tag (WSHPQFEK) fused to the C-terminus. This sequence was synthesized by gene synthesis (Twist Biosciences) and inserted between a KpnI site located immediately after the last codon of the Trx-thrombin coding cassette in pET32a and an XhoI site. The construct was verified through Sanger DNA sequencing, covering the entire coding region. The details concerning expression, protein refolding from insoluble inclusion bodies, and purification steps have been recently described[68].

## Expression and purification of InhA

InhA heterodimer was expressed as a secreted protein in suspension cultured expi293F cells with expi293™ expression medium. A heterodimer of full pro-inhibin was purified using in tandem -His and -Strep tags present on separate chains. Mature dimer composed of residues 311–426 of the inhibin βA subunit NP_002183.1 and 233–366 of the inhibin α subunit NP_002182.1, was further purified using a 10-mm C18 RPC column equilibrated in 5% buffer B (buffer A H$_2$O + 0.1% TFA; buffer B: CH$_3$CN + 0.1% TFA) with a linear gradient to 70% buffer B. Fractions containing InhA were pooled together, lyophilized, and reconstituted in PBS. Protein activity was determined by cell activity assay as previously reported[27].

## Protein folding assessment by NMR

Correct folding of selected mutant proteins was assessed by recording NMR natural abundance methyl-TROSY $^1$H-$^{13}$C correlation spectra (in case of BG$_{ZP-C}$) or 1D proton spectra (in case of zfBG$_O$) and by comparing the spectral fingerprints to the wild type proteins. NMR spectra were recorded using protein samples concentrated to at least 150 μM in 25 mM phosphate buffer pH 6.0 and using a band-selective optimized-flip-angle short-transient experiment[69] (SOFAST-HMQC) or 1D proton Bruker experiment with Watergate water suppression sequence. All measurements were recorded at 37 °C using a 600 or 700 MHz Bruker spectrometer equipped with Bruker TCI triple-resonance cryogenically cooled probes running Topspin v2.0 or v3.5. Data were processed with NMRPipe[70] and analyzed with the CcpNmr Analysis software package[71].

## NMR methyl spectra of BG$_{ZP-C}$: Growth Factor binding

Samples of $^2$H, $^{13}$C-methyl Isoleucine, Leucine, Valine (ILV) labeled BG$_{ZP-C}$ were prepared in 25 mM $^2$H-glycine buffer at pH 11.0 or 25 mM phosphate buffer at pH 6.0 in either straight 3 mm tubes (Wilmad, Vineland, NJ) or 5 mm susceptibility-matched microcells (Shigemi, Sigma-Aldrich). Separate samples, each with the labeled protein being 15–25 μM, were used for the titration. Spectra were recorded at 37 °C using a band-selective optimized-flip-angle short-transient experiment (SOFAST-HMQC)[69]. All measurements were recorded at 37 °C using a 600 MHz Bruker spectrometer equipped with a Bruker TCI triple-resonance cryogenically cooled probe running Topspin v3.5. Data were processed with NMRPipe[70] and analyzed with the CcpNmr Analysis software package[71].

## I161 methyl group assignment

To assign the chemical shifts of the δ1 methyl group of I161, we employed site-directed mutagenesis to substitute I161 with alanine. NMR spectra were recorded for both the wild type and mutant 150 μM samples in 25 mM phosphate buffer pH 6.0 using a band-selective optimized-flip-angle short-transient experiment (SOFAST-HMQC)[69]. The disappearance of a signal corresponding to the δ1 methyl group of I161 in the mutant spectrum allowed us to confidently assign the chemical shifts in the wild type protein.

## TGF-β2 biotinylation for SPR

TGF-β2 and its variants were dialyzed into 10 mM acetic acid and 125 μg aliquots were transferred into Eppendorf tubes and lyophilized overnight. Samples were reconstituted in 4 mM HCl. To remove traces of acetic acid, lyophilization was repeated and samples were resuspended in 50 μl of 4 mM HCl. 300 μl of 25 mM MES, pH 6.0. + 40% DMSO was added, followed by 0.0115 mg of ethyl-3-(3-dimethylaminopropyl)carbodiimide hydrochloride (EDC, Pierce) (60 nmol), 0.026 mg of sulfo-N-hydroxysulfosuccinimide (12 nmol, Pierce), and 0.20 mg (+)-biotinyl-3,6,9-trioxaundecanediamine (EZ-Link Amine-PEG3-Biotin, 480 nmol, Pierce). The reaction was allowed to proceed for 2 h at room temperature and then quenched by the addition of 1 ml 100 mM acetic acid. The growth factor was repurified by loading it onto a MonoS high resolution cation exchange column equilibrated in 25 mM sodium acetate, 30% isopropanol pH 4.5 and eluting with salt using gradient from 0–0.7 M over 10 column volumes. Modification of the ligand was confirmed by LC-ESI-TOF MS.

## SPR measurements of BG$_{ZP-C}$ mutants

Binding studies of BG$_{ZP-C}$ or zfBG$_O$ mutants were performed using streptavidin-coated sensor chips for capture of biotinylated ligands. The surface of a CM-5 sensor chip (GE Healthcare) was activated with EDC and N-hydroxysulfosuccinimide (NHS) (Pierce) followed by injection of streptavidin (Pierce) diluted into sodium acetate at pH 4.5 until the surface density reached 6000–8000 RUs. Biotinylated ligands were captured onto the streptavidin surface to a surface density between 50–300 RU. Equilibrium binding assays were performed by injecting the analytes in 10 mM HEPES, 150 mM NaCl, 0.1% surfactant P20 (Pierce) at pH 7.4 (HBS-EP buffer) at a rate of 10 μl min$^{-1}$ for 720 s, followed by a dissociation period of 600 s. Regeneration of the surface was achieved by a 10 s injection at 100 μl min$^{-1}$ of 4 M guanidine hydrochloride in 4 mM HCl solution. Baseline correction was performed by double referencing. The data were analyzed by fitting the results to a 1:1 kinetic model using the SPR analysis software Scrubber (BioLogic Software).

## SPR measurements of zfBG$_O$ mutants

SPR experiments were performed with a BIAcore X100 system (Cytiva). Neutravidin was coupled to the surface of a CM5 chip (Cytiva) by EDC-NHS activation of the chip, followed by injection of neutravidin (Thermo) over the surface in sodium acetate, pH 4.5 until the RU increased by 10000-15000 RU. Minimally biotinylated TGF-β2 was prepared by NHS/EDC activation followed by addition of EZ-Link Amine-PEG3-Biotin (Thermo). After removal of reagents, biotin-TGF-β2 was captured onto the chip surface at a maximum density of 50 RU. All experiments were performed in 10 mM CHES, 150 mM NaCl, 3 mM EDTA, 0.01% P20 surfactant, pH 8.6 at an injection rate of 100 μl min$^{-1}$. The higher pH was used to minimize non-specific interactions with the chip surface. The surface was regenerated in between each injection with a 10 second injection of 0.2 M guanidine hydrochloride. The experimental sensorgrams were obtained with double referencing with a control surface coated similarly with neutravidin but lacking the captured TGF-β2 and 8 blank buffer injections at the beginning of the run before injection of the samples. The data were analyzed by fitting the results to a 1:1 kinetic model using the SPR analysis software Scrubber (BioLogic Software).

## SPR measurements of BG$_O$ potentiation of TGFBR2 binding

SPR binding studies were performed on BIAcore T200 (Cytiva) and analyzed with the Biacore T200 analysis Software v3.0 at the Francis Crick Institute. TGF-β2 was coupled to the surface of a CM4 chip (Cytiva) by EDC-NHS activation of the chip, followed by injection of 1 μM TGF-β2 over the surface in sodium acetate, pH 4.5 until the RU increased by maximum 400 RU. All experiments were performed in 10 mM CHES, 150 mM NaCl, 3 mM EDTA, 0.01% P20 surfactant, pH 8.0. The higher pH was used to minimize non-specific interactions with the chip surface. The A-B-A injection scheme was applied at an injection rate of 30 μl min$^{-1}$ in sequence 60-90-60 s. In A-B-A mode, buffer A with constant concentration of BG$_O$ was first injected over the sensor chip surface to establish baseline. This was followed by the injection of buffer B, composed of buffer A with added analyte. After the buffer B injection, another injection of buffer A was performed. The surface was regenerated in between each injection with a 10 second injection of 0.2 M guanidine hydrochloride pH 2.5. The experimental sensorgrams were obtained with double referencing with a control surface and 4 blank buffer injections. The data were analyzed by fitting the results to a 1:1 kinetic model using the SPR analysis evaluation tool (Cytiva).

## SPR measurements of additional mutants of BG$_{ZP-C}$ (N82Y, I161Y, A158E) and TGF-β2 mutants (Q57A, K60A, L64A and Y50A, L51A)

SPR binding studies were performed on BIAcore T200 (Cytiva) and analyzed with the BIAcore T200 analysis Software v3.0 at the Francis Crick Institute. TGF-β2 or TGF-β2 mutants (Q57A, K60A, L64A and Y50A, L51A) were coupled to the surface of a CM4 chip (Cytiva) by EDC-NHS activation of the chip, followed by injection of 1 μM ligand over the surface in sodium acetate, pH 4.5 until the RU increased by maximum 400 RU. All experiments were performed in 10 mM CHES, 150 mM NaCl, 3 mM EDTA, 0.01% P20 surfactant, pH 8.0. injection rate of 100 μl min$^{-1}$. The surface was regenerated in between each injection with a 10 s injection of 0.2 M guanidine hydrochloride. The experimental sensorgrams were obtained with double referencing with a control surface and 4 blank buffer injections. The data were analyzed by fitting the results to a 1:1 kinetic model with drift using the SPR analysis evaluation tool (Cytiva).

## Luciferase reporter assays

L6E9 cells were seeded at a density of 20,000 cells/well in a 48-well plate. The following day, cells were transfected with the CAGA$_{12}$-luciferase reporter (200 ng/well) and either empty vector or a BG$_{ZP-C}$ construct (100 ng/well) using Lipofectamine 3000, following the manufacturer's protocol. Twenty-four hours after transfection, cells were serum starved for an additional 24 h. After starvation, cells were stimulated or not with TGF-β2 (1 pM) for 6 h. Cells were then lysed in 50 μl/well passive lysis buffer (25 mM Tris-phosphate [pH 7.8], 10% [v/v] glycerol, 1% [v/v] Triton X-100, 1 mg/ml bovine serum albumin, 2 mM EDTA) for 10 min at room temperature with agitation. Twenty microliters of cell lysis supernatant were combined with 100 μl of assay buffer (15 mM potassium phosphate [pH 7.8], 25 mM glycylglycine, 15 mM MgSO$_4$, 4 mM EDTA, 2 mM adenosine triphosphate, 1 mM dithiothreitol, 0.04 mM D-luciferin), and luciferase activity was measured on an Orion II microplate luminometer (Berthold Detection Systems).

## Crystallization, structure determination and refinement

Automated screening for crystallization was carried out using the sitting drop vapor-diffusion method with an Art Robbins Instruments Phoenix system in the Structural Biology Core at The University of Texas Health Science Center at San Antonio. Crystals of mmTGF-β2:BG$_{ZP-C}$ were grown at 22 °C in 13% PEG 4000, 0.1 M sodium citrate pH 5.6 and 10% ethylene glycol mixed in a 1:1 ratio for a total drop volume of 0.4 μL. Likewise, crystals of TGF-β3WD:BG$_O$:TGFBR1:TGFBR2 were grown at 22 °C in 30% PEG 4000, 0.1 M Tris HCl pH 8.5 and 0.2 M lithium sulfate. Diffraction data were collected at the Advanced Photon Source, Argonne, IL, NE-CAT beamlines for crystals flash-cooled in liquid nitrogen after wicking off excess solution from the crystals harvested in nylon cryo-loops without the use of additional cryoprotectant. Data were processed using AUTOPROC[72]. The structure of mmTGF-β2:BG$_{ZP-C}$ was determined by the molecular replacement method implemented in PHASER[73] using coordinates from PDB entries 3QW9 (ref. [33]) and 5TX6 (ref. [38]) as search models. The structure of TGF-β3WD:BG$_O$:TGFBR1:TGFBR2 was determined using coordinates from PDB entries 2PJY (ref. [42]) and 6MZN (ref. [21]) as search models. A successful molecular replacement phase solution was achieved for this complex when searching with separate OD1 and OD2 subdomains for the BG$_O$ component. All coordinates were refined using Phenix[74] with simulated annealing and TLS refinement, and alternated with manual rebuilding steps using Coot[75]. The models were verified using composite omit map analysis[76]. Data collection and refinement statistics are shown in Supplementary Table 1.

## Cryo-EM analysis of the TGF-β1:zfBGo:(TGFBR2)$_2$ complex and model refinement

Aliquots of 3 μl of each complex were applied to glow discharged UltrAuFoil r1.2/1.3 grids. The grids were blotted for 3 s at 100% humidity with force "2" and plunge frozen into liquid ethane using a Vitrobot Mark IV (Thermo Fisher). 41,335 movies were recorded on a 300 kV Titan Krios microscope (Thermo Fisher) equipped with a FalconIV camera and a Selectris energy filter using 0.72 Å/pix. The total

dose was set to 50 electrons per Å[2]. Movies were corrected using patch motion correction in CryoSPARC[77]. The contrast transfer functions (CTFs) were determined using patch CTF and an initial stack of particles were picked using Blob picker followed by template picker after initial reconstruction. 18,265,163 particles were extracted from micrographs and were subjected to several rounds of 2D classifications followed by ab-initio reconstructions with subsequent heterogenous refinements in CryoSPARC[77] that provided a final set of 307,870 particles. Bayesian polished particles using RELION-4.0 (ref. [78]) resulted in reconstruction that was refined using a non-uniform refinement function to 3.72 Å, with significant preferred orientation (cFAR = 0.01; SCF = 0.42). The resolution metrics provided here follow the gold-standard Fourier shell correlation 0.5 criterion (we adhered to a more stringent FSC criterion when determining the reported resolution due to preferred orientation). For illustration purposes and to aid model building, the cryo-EM map was processed with EMReady[79] or blurred using gaussian filter in ChimeraX[80]. The individual component models from the crystal structure (PDB: 9B9F - this study, 3KFD) were docked into the map using UCSF ChimeraX[80] followed by flexible fitting with Namdinator[81] with default parameters. The model was improved using iteratively Phenix real space refinement,[74] with manual building in Coot[75]. The analysis workflow is illustrated schematically in Supplementary Fig. 10.

### Cryo-EM analysis of the TGF-β3WD:zfBGo:TGFBR1:TGFBR2 complex and model refinement

Aliquots of 3 µl of each complex were applied to glow discharged UltrAuFoil r1.2/1.3 grids. The grids were blotted for 3 s at 100% humidity with force "2" and plunge frozen into liquid ethane using a Vitrobot Mark IV (Thermo Fisher). 7376 movies were recorded on a 300 kV Titan Krios microscope (Thermo Fisher) equipped with a FalconIV camera and a Selectris energy filter using 0.83 Å/pix. The total dose was set to 50 electrons per Å[2]. Movies were corrected using patch motion correction in CryoSPARC[77]. The contrast transfer functions (CTFs) of the flattened micrographs were determined using patch CTF and an initial stack of particles were picked using Blob picker followed by template picker after initial reconstruction. 3,001,720 particles were extracted from micrographs and were subjected to several rounds of 2D classifications followed by ab-initio reconstructions with subsequent heterogenous refinements in CryoSPARC[77] that provided final set of 281,881 particles. Bayesian polished particles using RELION-4.0 (ref. [78]) resulted in reconstruction that was refined using non-uniform refinement function to 4.10 Å, with significant preferred orientation (cFAR = 0.05; SCF = 0.89). The resolution metrics provided here follow the gold-standard Fourier shell correlation 0.5 criterion (we adhered to a more stringent FSC criterion when determining the reported resolution due to preferred orientation). For illustration purposes and to aid model building, the cryo-EM map was processed with EMReady[79]. The crystal structure of the complex (PDB: 9B9F, this study)[82] was docked into the map using UCSF ChimeraX[80] followed by flexible fitting with Namdinator[81] with default parameters. The model was improved using iteratively using Phenix real space refinement,[74] with manual building in Coot[75]. The analysis workflow is illustrated schematically in Supplementary Fig. 10.

### Cryo-EM analysis of the TGF-β1:ratBG$_O$:(TGFBR2)$_2$ complex and model refinement

Aliquots of 3 µl of complex sample was applied on UltrAuFoil r1.2/1.3 grids covered with graphene oxide as described previously[83] and the grids were blotted for 3.5 s at force -1 at 4 °C and plunge frozen into liquid ethane using a Vitrobot Mark IV (Thermo Fisher). 26,386 movies were recorded on a 300 kV Titan Krios microscope (Thermo Fisher) equipped with a FalconIV camera and a Selectris energy filter using 0.921 Å/pix. The total dose was set to 50 electrons per Å[2].

Movies were corrected using motion correction as implemented incorporated into RELION_4.0 software package[78]. The contrast functions (CTFs) were determined using GCTF[84] and an initial stack of particles were picked using crYOLO with general model[85]. 1,016,267 particles extracted from micrographs using RELION-4.0 (ref. [78]) were imported into CryoSPARC[77] and subjected to 3 rounds of initial 2D classification and several rounds of ab-initio reconstructions with subsequent heterogenous refinements that provided 101,180 particles that were used to train TOPAZ[86] and subsequent particle picking. 3,978,615 picked particles were extracted from micrographs and were subjected to several rounds of 2D classifications followed by ab-initio reconstructions with subsequent hetero refinements in CryoSPARC[77] that provided final set of 230,626 particles. Bayesian polished particles resulted in reconstruction that was refined using non-uniform refinement function to an overall resolution 3.40 Å. The resolution metrics provided here follow the gold-standard Fourier shell correlation 0.143 criterion. For illustration purposes and to aid model building, the cryo-EM map was processed with EMReady[79]. The individual component models from AlphaFold2[82] were docked into the map using UCSF ChimeraX[80] followed by flexible fitting with Namdinator[81] with default parameters. The model was improved using iteratively Phenix real space refinement[74], with manual building in Coot[75]. The analysis workflow is illustrated schematically in Supplementary Fig. 10.

### Cryo-EM analysis of the TGF-β:BG complex

Aliquots of 3 µl of complex sample was applied on UltrAuFoil r1.2/1.3. The grids were blotted for 3 s at 100% humidity with force "2" and plunge frozen into liquid ethane using a Vitrobot Mark IV (Thermo Fisher). 9496 movies were recorded on a 300 keV Titan Krios microscope (Thermo Fisher) equipped with a FalconIV camera and a Selectris energy filter using 0.83 Å/pix. (including 2181 movies recorded on 20° tilted stage) The total dose was set to 55 electrons per Å[2]. Movies were corrected using patch motion correction in CryoSPARC[77]. The contrast functions (CTFs) were determined using patch CTF and an initial stack of particles were picked using Blob picker followed by template picker after initial reconstruction. 8,179,080 particles were extracted from micrographs and were subjected to several rounds of 2D classifications followed by ab-initio reconstructions with subsequent heterogenous refinements in CryoSPARC[77] that provided final set of 256,006 particles. Final reconstruction that was refined using non-uniform refinement function yielded a map at resolution 6.39 Å. The resolution metrics provided here follow the gold-standard Fourier shell correlation 0.5 criterion (we adhered to a more stringent FSC criterion when determining the reported resolution due to preferred orientation).The individual component models from the above structures or AlphaFold[82] were docked into the map using UCSF ChimeraX[80]. The analysis workflow is illustrated schematically in Supplementary Fig. 10.

### Coupling of O6-(4-Aminomethyl-benzyl)guanine (BG-NH2) to CF640R-NHS

The reaction vial was charged with CF640R-NHS (MW: 929.19, 1 µmol, 1 mg) dissolved in 300 µl of DMSO and BG-NH2 (MW: 270.3, 1.4 µmol, 0.4 mg) dissolved in 200 µl DMSO. A 3.0-fold molar excess of dry triethylamine (MW: 101.19, density: 0.726 g/ml, 0.326 mg -0.44 µl) was added to the reaction vial. The reaction was stirred under argon overnight at 30 °C. The reaction was monitored using LC-MS (Cal M.W: 1084.49, Found: 1084.82) and was directly purified on a PerkinElmer HPLC system using solvent A (0.08%TFA and 0.1% acetonitrile in water) and solvent B (0.1% TFA in acetonitrile) as an eluent in gradient 0–40% over 40 min to give the conjugated compound. The synthesized ligand, hereafter referred to as 'SNAP-CF640R' was dissolved in DMSO to prepare a 1 mM concentration stock.

## Establishment of a stable HEK293T SNAP-BG cell line

To establish a stable HEK293T cell line expressing the SNAP-BG fusion protein, the pCDNA3.1+ plasmid containing the SNAP-BG fusion gene was linearized using the FspI restriction enzyme. The linearized plasmid was then transfected into HEK293T cells at 80% confluency using FuGENE HD reagent following the manufacturer's protocol (Promega, Madison, WI, USA). Following transfection, positive selection was initiated by the addition of hygromycin at a concentration of 400 μg/ml. A pool of cells expressing different levels of SNAP-BG was selected for further study.

## Evaluation of SNAP-BG expression, localization and activity

To assess SNAP-BG expression and correct subcellular localization, cells were grown on glass coverslips until reaching approximately 70% confluency. Subsequently, the cells were incubated with 30 nM SNAP-CF640R ligand for 20 minutes. After ligand incubation, excess dye was removed by washing three times with PBS and the cells were fixed with 4% paraformaldehyde (PFA), stained with DAPI and washed again twice with PBS. Coverslips with fixed cells were mounted onto glass slides using mounting medium (ProLong™ antifade reagent) and subjected to confocal microscopy analysis using a Leica SP8 confocal microscope.

The activity of SNAP-BG was assessed as previously reported[19]. Briefly, HEK293T cells expressing SNAP-BG were treated with low concentration (20, 40 pM) of TGF-β2 for 15 min. Cells were then washed, lysed using lysis buffer and assayed for the presence of pSMAD2 using Western Blot by probing the membrane with anti-pSMAD2 antibody (RRID: AB_490941) followed by HRP-conjugated anti-rabbit secondary (RRID: AB_2617138). The presence of SNAP-BG was also confirmed using Western Blot by probing the membrane with anti-SNAP-tag antibody (RRID: AB_10710011) or anti-Betaglycan antibody (RRID: AB_2202608) followed by HRP-conjugated anti-rabbit or anti-goat secondary antibodies (RRIDs: AB_2617138 or AB_2617143, respectively).

## Fluorescent labeling of TGF-β2 and InhA

Purified TGF-β2 (0.5 mg) or InhA (0.1 mg) at a concentration of 2 mg/ml in 10 mM acetic acid was combined with 100 μl of PBS pH 7.2 and 60 μl of DMSO. The pH was adjusted to approximately 7.0. A 5:1 molar excess of the succinimide ester of CF640R fluorescent dye (Biotium) or Alexa Fluor™ 647 was then added to the protein solution. The labeling reaction was allowed to proceed for 60 min and was quenched by adding 100 μl of 100 mM Tris-HCl pH 8.0. The labeled protein was then dialyzed against 10 mM acetic acid and purified using a cation exchange MonoS column (Cytiva) with a gradient of NaCl. Fractions containing the labeled protein were identified using gel electrophoresis and the ImageQuant™ 800 fluorescent gel documentation system (Cytiva). Pooled fractions containing the labeled protein were dialyzed against 50 mM acetic acid, concentrated, and stored at -20 °C until further use.

## Competition binding assay

HEK293T cells stably expressing either the SNAP-BG construct or a control strain were seeded onto 18 mm coverslips placed in 12-well plates at a density of $10^5$ cells per well. Upon reaching approximately 80% confluency, cells were treated with 20 ng ml⁻¹ of fluorescently labeled TGF-β2 or 20 ng ml⁻¹ InhA. After 10 min, a 100-fold excess of the indicated unlabeled ligand was added, and cells were further incubated for 15 min. Subsequently, SNAP-CF568 at final concentration of 30 nM was introduced, and cells were incubated for an additional 15 min. Excess dye was removed by washing the cells three times with PBS. Cells were then fixed with 4% paraformaldehyde (PFA), stained with DAPI and washed again 2X with PBS. Fixed cells on coverslips were carefully mounted onto glass slides using mounting medium (ProLong™ antifade reagent). Subsequently, the samples were subjected to detailed confocal microscopy analysis using a Leica SP8 confocal microscope.

## Confocal microscopy and image processing

The samples were subjected to confocal microscopy analysis using a Leica SP8 confocal microscope using an HC PL APO CS2 20x/ 0.75 IMM objective at 25 °C with the following confocal settings: pinhole 1 airy unit, scan speed 400 Hz unidirectional, format 1024 ×1024 pixels at 16 bit. Images were collected using hybrid detectors and an argon, 561 nm and 633 nm lasers with 2x line averaging. For each field of view (FOV), a stack of 25 images was collected with a z-step of 0.5 μm. Collected images were processed using Fiji software[87]. Images from the stack were combined into a single z-projection, and a threshold was established using the Otsu method for automatic thresholding, using the SNAP-BG signal channel as source. The intensity of the defined areas was measured for both the SNAP-BG and TGF-β2/inhibinA channels. The ratio of intensities for each channel was calculated and plotted for visualization using scatter plots in GraphPad Prism version 10.1.1 for Mac, GraphPad Software, Boston, Massachusetts USA. Each data point represents the ratio of SNAP-BG signal to ligand signal within a segmented region of the image generated by Otsu thresholding.

## Alphafold2-multimer simulations

The interactions between TGF-β family members growth factors and betaglycan subdomains $BG_{ZP-C}$ and $BG_O$ were predicted using the AlphaFold2 multimer protocol. Protein sequences (dimer for Growth factor + monomer $BG_{ZP-C}$ or $BG_O$) were used as an input. In total, 25 models per complex were computed. The predicted structures were evaluated based on the pTM+ipTM score. The pTM score (predicted TM score) estimates how well the predicted structure matches the true structure, based on template modeling principles, whereas ipTM (interface predicted TM score) measures the confidence of the predicted interface between the interacting proteins. Both pTM and iPTM range from 0 to 1.

## Assessment of BG complex formation with pro-TGF-β2

50 μM samples of $BG_O$ or BG were mixed with pro-TGF-β2 at a 1:1 ratio. Mixtures were incubated at room temperature for 15 minutes to allow potential interactions. Following incubation, the mixtures or separate proteins were loaded onto a size-exclusion chromatography (SEC) column (Superdex 200 Increase 10/300 GL). The column was pre-equilibrated with PBS buffer, pH 7.4 and the chromatography run was carried out at a flow rate of 0.5 ml/min. 0.3 ml Fractions corresponding to specific elution peaks were analyzed by SDS-PAGE to assess the composition of each fraction.

## Statistics and reproducibility

With the exception of the structure determinations, all experiments were repeated at least three times. Statistical analysis used in Fig. 3f was 2-way Anova.

## Reporting summary

Further information on research design is available in the Nature Portfolio Reporting Summary linked to this article.

# Data availability

All coordinates, structure factors and Cryo-EM maps have been deposited in the Protein Data Bank (PDB) with the following accession numbers: 8DC0 (crystal structure of $BG_{ZP-C}$: mmTGF-β2 complex), 9B9F (Crystal structure of TGF-β3WD:$BG_O$:TGFBR1:TGFBR2 complex), 9FK5 (Cryo-EM structure of TGF-β3WD:zf$BG_O$:TGFBR1:TGFBR2 complex), 9FKP (Cryo-EM structure of TGF-β1:zf$BG_O$:(TGFBR2)$_2$ complex), and 9FDY (Cryo-EM structure of TGF-β1:rat$BG_O$:(TGFBR2)$_2$ complex). Additionally, Cryo-EM maps were deposited int the Electron

Microscopy Data Bank (EMDB) with the following accession numbers: EMD-50519 (Cryo-EM structure of TGF-β3WD:zfBG$_O$:TGFBR1:TGFBR2 complex), EMD-50524 (Cryo-EM structure of TGF-β1:zfBG$_O$:(TGFBR2)$_2$ complex), EMD-50333 (Cryo-EM structure of TGF-β1:ratBG$_O$:(TGFBR2)$_2$ complex) and EMD-50326 (Cryo-EM map of TGF-β1:BG complex). Plasmids generated in this study are maintained in the laboratories of Andrew Hinck (ahinck@pitt.edu) and Caroline Hill (caroline.hill@crick.ac.uk) and will be made available upon request. Source data are provided with this paper.

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

## Acknowledgements

We would like to thank Simone Kunzleman and Chloe Roustan from the Structural Biology STP at the Crick for assistance with SPR experiments and HEK293T protein expression and Hema Nagaraj from the Chemical Biology STP for providing fluorescently-labeled SNAP-ligand. We would also like to thank Mateo Coppola who helped with the production of the BG$_O$ mutants, Jue Wang from DeepMind for useful discussion, Scott Wilcockson for his helpful comments and corrections, James Conway from the Cryo-EM laboratory at the University of Pittsburgh and Daniel Clare from UK national electron Bio-Imaging Centre for support in microscope alignment and data collection. This research was supported by the NIH through R01 grants (GM58670 and CA233622) awarded to A.P.H. and the CIHR through a grant (PJT-191766) awarded to D.J.B. This work was also supported by the Francis Crick Institute, which receives its core funding from Cancer Research UK (CC2021, CC2058), the UK Medical Research Council (CC2021, CC2058), and the Wellcome Trust (CC2021, CC2058) to C.S.H. and P.C., respectively. This project additionally received funding from the European Union's Horizon 2020 research and innovation programme under the Marie Skłodowska-Curie Grant Agreement No. 893196 to L.W. This work is based upon research conducted in the Structural Biology Core, a part of the Institutional Research Cores at the University of Texas Health Science Center at San Antonio supported by the Office of the Vice President for Research, Greehey Children's Cancer Research Institute, and the Mays Cancer Center Drug Discovery and Structural Biology Shared Resource (NIH P30 CA054174). The Rigaku HyPix-6000HE Detector, Universal Goniometer and VariMax-VHF Optic instrumentation are funded by NIH-ORIP SIG Grant S10 OD030374. X-ray data were collected at Northeastern Collaborative Access Team (NECAT) 24-ID beamline at the Advanced Photon Source, Argonne National Laboratory supported by the NIH (GM124165, RR029205, and OD021527) and the DOE (DE-AC02-06CH11357). Cryo-EM data was collected at the Cryo-EM laboratory at the University of Pittsburgh and at the UK national electron Bio-Imaging Centre (eBIC) under proposal BI37190. The Pittsburgh Center for Cryo-EM (RRID:SCR_025216) used for data collection in this project was supported, in part, by the University of Pittsburgh, the School of Medicine, the Department of Structural Biology, and the National Institutes of Health (grants S10-OD-019995 and S10-OD-025009). The content is solely the responsibility of the authors and does not necessarily represent the official views of the National Institutes of Health.

## Author contributions

L.W., C.S.H, and A.P.H. designed the study. L.W. and A.P.H. designed and executed experiments. A.B.T. determined the crystal structures. E.P. provided technical support in graphene grid preparation and sample freezing. L.W., J.A.C., and P.C. analyzed the Cryo-EM data. I.O.C., C-H.B., and R.I., provided support in SPR data acquisition, Cy.S.H. and T.K. provided support in protein expression. Y-F.L. and D.J.B. provided and analyzed signaling activity data. F.L-C. provided supervision. L.W., C.S.H., and A.P.H. wrote the manuscript. L.W., D.J.B., P.C., C.S.H. and A.P.H. provided funding for the study. Note that C.S.H. refers to Caroline S. Hill and Cy.S.H. refers to Cynthia S. Hinck.

## Funding

## Competing interests

The authors declare no competing interests.
