## [Transparent Peer Review file · Nature Communications]

Structures of TGF- β with betaglycan and signaling receptors reveal mechanisms of complex assembly and signaling

Corresponding Author: Dr Caroline Hill

Version 0:

Reviewer comments:

Reviewer #1

(Remarks to the Author)

Secreted BMP/GDF/TGF-beta proteins are essential for multicellular life. The authors used a comprehensive range of biophysical (crystallography, cryo-EM, NMR spectroscopy, and SPR), cellular, and computational experiments to elucidate the molecular mechanisms of TGF-beta signalling control by the transmembrane co-receptor Betaglycan (BG). They determined multiple crystal and cryo-EM structures of TGF-beta in complex with its receptors, as well as BG. These structures were used to guide site-directed mutagenesis coupled with biophysical (SPR-/NMR-based) binding and cellular signalling assays, illuminating key amino acid residues mediating TGF-beta interactions with BG and its receptors. This thorough, high-quality study of challenging-to-produce ligands and receptors significantly advances our understanding of BG-TGF-beta signalling. Therefore, this fundamental study could be published essentially as it stands.

I have several minor comments-questions (that do not require any further experiments) listed below.

1. Lines 160-163. The authors write, 'We hypothesized that the expanded density in the areas attributed to TGFBR2 occurred due to a large-scale closed-to-open transition of TGF- β 3 (Fig. 1E) 40, which likely hindered high-resolution reconstruction and confounded efforts to obtain crystals'. The cryo-EM map shown in Fig. 1D appears to be somewhat streaky. For discussion, please refer to Scheres 2016 in Methods in Enzymology:

'One problem that may arise at this point is that there are not enough different views for 3D reconstruction because the particles adopted a strongly preferred orientation on the experimental support. This problem, which is often already detectable from a shortage of different 2D class averages, may manifest itself in streaky reconstructions, where densities are smeared out in the direction of the predominant view. A related problem may be that different classes become streaky in different directions, which is an indication that the classification converged to separate different views rather than different structural states.'

Are the authors confident that the map features presented in Fig. 1D result from 'a large-scale closed-to-open transition of TGF- β 3' and not from the preferred particle distribution on cryo-EM grids as illustrated in EDF1 A-C? Would they consider rephrasing the sentence in lines 160-163 to reflect the possibility that the preferred particle orientation problem might have affected the cryo-EM maps?

2. Fig. 4A 1-2. Several dashed lines seem to imply hydrophobic interactions (W32-A249, W32-F250, W30-F250, V98-F49), but the selected atoms appear to be rather far apart (4-5 Å). Was any distance cut-off used to select the atoms connected by dashed lines? Similarly, do the side chains of D88 and R94 interact directly?

3. Have the authors considered depositing selected AlphaFold models (e.g. with iptm+ptm scores higher than 0.5 and/or the one discussed in more detail in the manuscript — three TGF-beta isoforms plus BG/InhA) as supplementary information files (in PDB format)?

4. Lines 509-513. Given that the authors explicitly discuss pH-dependent interactions between BMPs/GDFs and RGMs, wouldn't it be appropriate to consider citing two articles that demonstrate these pH-dependent interactions between BMP2/GDF5 and RGMs (PMID 25938661 and 32576689)? Full disclosure: this reviewer is the first author of PMID 32576689.

5. Line 600. Change RO1 to R01.
6. Line 638. Change '30..853' to '30–853'.
7. Line 660. In 'rat BGO', should 'O' be in subscript?
8. Line 662. Please provide more details on the BG(ZP-C) construct (amino acid residue range, linkers, protease cleavage sites, etc.), its expression (duration, temperature, etc.), and purification (buffers and purification columns). Ideally, please include the full-length amino acid sequence of the construct in the supplementary information file.
9. Lines 669-674. Please provide more details on the Inhibin A construct (amino acid residue ranges, secretion signal, etc.), its expression (duration, temperature, etc.) and purification (all buffers and purification columns). Ideally, please include the full-length amino acid sequence of the construct in the supplementary information file.
10. Lines 728-756. Why were the studies performed in 10 mM CHES at pH 8.0-8.6? Are BG(O) proteins more soluble or stable in this buffer? If so, it might be useful to indicate this.
11. Line 750. What is 'the A-B-A injection scheme'? Please clarify.
12. Lines 784-804. How/were crystals cryoprotected? If so, please indicate the concentration of cryoprotectants.
13. Line 810. Change 'keV' to 'kV'. Also, make this change in lines 834, 857, and 882.
14. Lines 813 and 885. Change 'contrast functions' to 'contrast transfer functions'.
15. Lines 806-828. Please cite papers describing CryoSPARC (which version was used?) and RELION-4.0.
16. Line 839. Change 'Particles' to 'particles'.
17. Multiple places. Why do certain words start with a capital letter in the middle of a sentence? For example, 'Grids' (line 854) and 'Movies' (lines 857 and 881). It should be RELION-4.0, not Relion 4.0: <https://relion.readthedocs.io/en/release-4.0/>
18. Figure 3 legend. Change 'Detailles' to 'Details'.
19. Figure 4D legend. Change 'into the binding interface' to 'to the binding interface'? Change 'PISAE PDB' to 'PDBE PISA': <https://www.ebi.ac.uk/pdbe/pisa/>
20. Figure 4D legend. 'Growth Factors from TGF-beta family'. Were BMPs/GDFs included in this analysis? If so, shouldn't it be 'Growth Factors from the TGF-beta superfamily'?
21. Sensorgrams in all main figures. Wouldn't it be easier for the reader to see the Kds in these figures straight away?
22. Fig. 5E. It is not clear what the yellow dashed lines indicate. Some residues appear to be too far apart to form hydrophobic interactions (e.g., I39 and Y248). It is difficult to determine without further information (was any distance cut-off used?).
23. Fig. 6D. Change 'Instensity' to 'Intensity'.
24. Fig. 7 legend. Both British and American English are used, such as 'signaling' and 'signalling'.
25. PDB ID 8DC0. Water molecules (their oxygen atoms) 62 and 76 clash with neighbouring protein oxygen atoms (distances of 1.96 and 1.91 angstroms, respectively). This likely contributes to the relatively high clashscore (7.88). I used Coot-Validate-Probe clashes. These are the top four clash hits.

Tomas Malinauskas (University of Oxford, UK)

Reviewer #2

(Remarks to the Author)

This work deals with the cell surface glycoprotein betaglycan (BG), and in particular on its interaction with different binding partners, from a structural focus with the aim of understanding its mechanisms of action. Due to my background I'll focus on the NMR experiments.

NMR is used with two main purposes here:

- to demonstrate that the mutations done in different domains of BG (BG<ZP-Csub> and zfBG<osub>) to test their impact, and thus relevance, in protein binding partners interactions, do not perturb BG domain structure.
- to monitor the interaction in solution of BG<ZP-Csub> with three protein ligands: TGF- β 2, mmTGF- β 2, and Inhibin A.

With respect to to the first issue:

- in page 11, line 291: "we recorded NMR natural abundance 2D 1H-13C shift correlation": 2D 1H-13C shift correlation is not any type of NMR spectra, it could be better "2D 1H-13C correlation spectra" or "2D 1H-13C HMQC spectra".

"... and 1D 1H spectra blabla" should be removed.

The conclusion of this paragraph should be less categorical: it is true that the NMR fingerprints of the mutants and wild type are very similar, but still there are differences, and we cannot assure that they do not arise from small changes in the structure. It should be better said that the mutations did not cause important changes in the overall protein structure.

- there is some confusion with the mutations at zfBG<osub>: in the manuscript is says N253A but in extended figure 4 it says D253A.

Second issue:

- In Figure 6A-C, ppm of both axes should be added.

In page 15, line 415: "Upon titration of 13C-methyl-Ile BGZP-C with InhA, we observed a concentration dependent weakening of signal intensities and peak broadening, which is indicative of binding (Fig. 6C-E)." Well, not necessarily, it could just mean BG aggregation or oligomerization? I would use "suggesting" rather than "indicating". Interestingly, this domain contains 6 Ile, as in fact reflected in the MeTROSY spectrum. Interestingly, all these Ile, except I102 gather in the same region in the 3D structure. Could it be that the InhA binding region is far away from any Ile residue (or in other words, the probe used here, which are Ile, are not good for detecting binding to InhA). In fact, the fact that I161 does not move indicates that the conformational change (from random coil to helix) observed for TGF- β binding does not happen upon InhA binding. It looks indeed that binding is happening, maybe accompanied with some oligomerization process (that's why the broadening)?, maybe at a different region?

Reviewer #3

(Remarks to the Author)

From a general perspective, I think that the work is highly interesting and provides new data that allow advancing the current knowledge in TFG recognition and signalling. The methodological approach used is properly designed with the combination of x-ray, cryo EM and NMR.

First, I would like to clarify that my specific comments are focus just on the NMR experiments.

In this context, some additional information should be provided to clarified certain points:

1) Regarding Extended Figure 4 and page 12 main text, where it is said: "Notably, the D253A mutant exhibits signal broadening across the spectra, hinting at more widespread structural changes that may suggest alterations in protein folding".

And page 12 where it is said: "One caveat to these results is that the N253A mutation appeared to perturb the structure".

The authors should further clarify this data interpretation since the signal broadening across the spectra could be also related to a change in the aggregation state of the protein. Then, further information should be required to asses that the D253A mutant displays a change in protein folding. For instance, a methyl-TROSY 1H-13C could be measured to confirm that there are chemical shift changes that correlate with a different protein folding and not only line broadening in this construct.

2) In Figure 6 the 1H and 13C scales of the NMR spectra are missing, the scales should be included. Moreover, the type of 1H-13C correlation experiment that has been used should be specify in the figure caption.

3) Finally, the authors should clarify the assignment protocol that has been carried out to identify the chemical shifts of delta1 methyl of Ile 161.

Reviewer #4

(Remarks to the Author)

The paper by Wieteska and colleagues presents complementary structural approaches (X-ray crystallography, NMR and cryo-EM) towards solving the structure of the co-receptor betaglycan (BG) with its ligands (TGF-beta1, -beta2, beta3, inhibin-A) and further even the high molecular complex of BG-ligands and signaling type II and type I receptors of TGF-beta. The paper comes as a natural sequel after a series of structural papers on BG generated by the AP Hinck lab. The unique contribution of the new paper is the ingenious bypass of several hurdles that prohibit the crystallization of large multi-domain parts of BG together with its ligands and receptors. The combination of clever mutant ligands, multiple ligands and BG from zebrafish and rat take the structural TGF-beta field one step further and allows the authors to present exciting new structural models of these important receptors bound to several of their ligands, thus advancing our understanding of signal transduction and refining older observations based on a solid perspective from the structural field.

The paper is succinctly written and provides a large resource of new structures. I have some conceptual comments that relate to the interpretation of the data and the signaling complex, few comments on missing controls and a general request on presentation of statistical details in the relevant data panels.

Comments:

1. To assist the non-specialist, figure 1 can start with a cartoon of one ligand and the three receptors, aiming at highlighting the relevant domains of each protein that are analyzed in the paper. This is most relevant for the BG domains: BGO, BGZP-N and BGZP-C.
2. Figure 2A: by comparing DAPI and BG staining, not all cells appear to express strong receptor levels in this stable clone of the cells. The TGF-beta2 staining pattern appears to overlap exactly with the BG pattern. Yet, can the fluorescent TGF-beta2 bind to endogenous (unlabeled) receptors? This does not complicate the co-localization analysis but is relevant to the signaling question presented below. Explain in the legend that this is rat BG.
3. Ext data Figure 2A: the SNAP-BG appears as a sharp band of 120 kDa. Does this BG become GAGylated generating a long smear on the gel? To get a feeling on the degree of overexpression, this western blot could be re-probed with a BG antibody that will also detect the endogenous human BG. Explain in the legend that this is rat BG.
4. Ext data Figure 2C: for how long were the cells exposed to TGF-beta2? In this experiment, the signaling in the minus SNAP-BG lanes is mediated by the endogenous BG, right? This is why in the previous comment, monitoring the endogenous BG was suggested.
5. The discussion presents the possible scenario of BG internalization, low pH and dissociation of BG from the ligand-receptor complex. Can the SNAP-BG system also monitor BG internalization together with TGF-beta2? Do the two signals dissociate in acidic endosomes?
6. Figure 3E indicates similar KD values for TGF-beta2 binding to BGO and BGZP-C, around 100 nM. Figure 4C indicates KD value for TGF-beta2 binding to BGO, around 50 nM. Is it safe to assume that the 2 domains, BGO and BGZP-C, have equal affinity for the ligand? In the BG dissociation model, does the binding of type I receptor decrease the KD significantly? I wonder whether the KD for ligand binding of each domain is different when the domain is present in native BG and not isolated.
7. Figure 5A-C: is the difference between BG from zebrafish and rat conserved also in human BG?
8. The discussion and figure 7 present the model of TGF-beta signaling. This raises some fundamental questions. I recommend that the authors include some further clarifications and a short expansion of the discussion. Why is it not preferable that the complex shown in Figure 11 is the signaling complex? Is such a complex with BG and only one set of type II/type I receptors expected to signal "less" compared to a complex with two sets of II/I receptors? If there is evidence for this, it is worth presenting it in some detail in the discussion.
9. Continuing in the same line of discussion, the authors discuss the importance of receptor internalization and of receptor cleavage, which is relevant and appropriate. In the same context, it would be useful to discuss also the mechanism of ligand-coreceptor assembly from the latent form of the ligand. It is obvious that all solved BG structures use mature ligands or even mutants like the WD mutant used in this study for technical reasons. A brief discussion on whether the authors envision first the generation of a local pool of mature ligand from its latent depots, that then binds to BG with the calculated KD from the studies in vitro, and then the formation of the hyper-complex analyzed in this paper, will enlighten the reader.

Technical comments:

1. Figure 1C, F, H, K and other figures later: explain in the legend the importance of the 9 microphotographs. They are presented as corresponding 2D classes but the method presents millions or hundreds of thousands of classifications, so what do the 9 images represent, and why 9?
2. Figure 1F, H: explain in the legend the SEC peaks labeled as (b).
3. Ext data Figure 2D: explain in the legend what each data point represents. A single cell?
4. Figure 1B: explain in the legend what each data point represents. A single cell?
5. Figure 1C, D: explain in the legend the statistical parameters: standard deviation or SEM, number of repeats.
6. Figure 1C, D and later: include a method on AF2M and explain iptm and ptm.
7. Figure 3B, E, F (luciferase) and additional SPR assays in the other figures: explain in the legend the statistical parameters: standard deviation or SEM, number of repeats. In panel 3F, also explain the p-value of 4 stars.
8. Figure 4E: explain in the legend that the numbers on top of the brackets refer to the number of mutations. The quadruple mutant requires a bracket with a 4.
9. Ext data Figure 5D, E: in the legend, correct the panel labeling. First D and the last panel is E.
10. Figure 6D, E: explain in the legend the details of these diagrams: median line in panel D, quartiles in panel E?

Version 1:

Reviewer comments:

Reviewer #1

(Remarks to the Author)

The authors have fully addressed my concerns, and I fully support the publication of this work.

Reviewer #2

(Remarks to the Author)

The authors have successfully addressed the issues raised.

Reviewer #3

(Remarks to the Author)

- My concerns have been clarified and therefore, I recommend the publication of the manuscript.

Reviewer #4

(Remarks to the Author)

In their revised manuscript, Wieteska and colleagues have delivered a very clear and convincing account on the structural details of a signaling super-complex residing on the plasma membrane, the betaglycan-TGF-beta-TGF-beta receptor complex. This protein assembly is known for its powerful biological actions during embryonic development and in tissue homeostasis, while playing diverse roles in several human diseases. The contribution that this paper makes is significant as it bridges relatively fragmented previous knowledge into a more comprehensive model of assembly of the co-receptor and the receptors in the presence of extracellular ligands. The interest that this paper generates will find followers in the structural, cellular developmental and cancer biology fields.

Aristidis Moustakas

Rebuttal to reviewers' comments

Reviewer #1 (Remarks to the Author):

Secreted BMP/GDF/TGF-beta proteins are essential for multicellular life. The authors used a comprehensive range of biophysical (crystallography, cryo-EM, NMR spectroscopy, and SPR), cellular, and computational experiments to elucidate the molecular mechanisms of TGF-beta signalling control by the transmembrane co-receptor Betaglycan (BG). They determined multiple crystal and cryo-EM structures of TGF-beta in complex with its receptors, as well as BG. These structures were used to guide site-directed mutagenesis coupled with biophysical (SPR-/NMR-based) binding and cellular signalling assays, illuminating key amino acid residues mediating TGF-beta interactions with BG and its receptors. This thorough, high-quality study of challenging-to-produce ligands and receptors significantly advances our understanding of BG-TGF-beta signalling. Therefore, this fundamental study could be published essentially as it stands.

We thank the reviewer for their very positive comments about our work and are delighted to see that they consider that the study could be published essentially as it stands.

I have several minor comments-questions (that do not require any further experiments) listed below.

1. Lines 160-163. The authors write, 'We hypothesized that the expanded density in the areas attributed to TGFBR2 occurred due to a large-scale closed-to-open transition of TGF- β 3 (Fig. 1E) 40, which likely hindered high-resolution reconstruction and confounded efforts to obtain crystals'. The cryo-EM map shown in Fig. 1D appears to be somewhat streaky. For discussion, please refer to Scheres 2016 in Methods in Enzymology:

'One problem that may arise at this point is that there are not enough different views for 3D reconstruction because the particles adopted a strongly preferred orientation on the experimental support. This problem, which is often already detectable from a shortage of different 2D class averages, may manifest itself in streaky reconstructions, where densities are smeared out in the direction of the predominant view. A related problem may be that different classes become streaky in different directions, which is an indication that the classification converged to separate different views rather than different structural states.'

Are the authors confident that the map features presented in Fig. 1D result from 'a large-scale closed-to-open transition of TGF- β 3' and not from the preferred particle distribution on cryo-EM grids as illustrated in EDF1 A-C? Would they consider rephrasing the sentence in lines 160-163 to reflect the possibility that the preferred particle orientation problem might have affected the cryo-EM maps?

We thank the reviewer for highlighting this issue. We have now updated the manuscript to explicitly discuss the possibility that a strong preferred orientation of the particles contributes to the streakiness of the reconstruction and have referred to the Scheres 2016 paper, as suggested (**page 7 lines 160-166**).

While we initially considered preferred orientation, which is indeed severe for this complex (reproduced in Figure for Reviewer #1, panel A, below), as solely responsible for the greatly expanded density in our analysis, we also considered the possibility of a contribution from the previous well-characterized closed-to-open transition, which results in a large-scale rearrangement of the TGF- β protomers relative to one another. Therefore, we replaced TGF- β 3 with TGF- β 1 in the subsequent complex (original Figure 1G – now Figure 1H), given that TGF- β 1 is structurally similar to TGF- β 3, binds to the same set of receptors, and is essentially locked in the closed state, as previously shown (Huang, et. al, PMID: 21423151). As explained in the manuscript, this substitution significantly reduced the expanded density such that it much more closely encompassed the model, although notably the uneven distribution of the views

Figure for Reviewer #1: Particle distribution for presented maps on Figure 1E and Figure 1H for TGF- β 3:zfBG_o:(TGFBR2)₂ (A) and TGF- β 1:zfBG_o:(TGFBR2)₂ (B) complexes, respectively.

persisted (reproduced in Figure for Reviewer #1, panel B, above). For this reason, we think that the observed expansion of the density (Figure 1E), particularly in the region attributed to TGFBR2 most distant from the BG orphan domain, was mostly caused by the flexibility of the TGF- β 3 ligand, though we fully concede preferential alignment also contributed to the less-than-ideal streaky density.

2. Fig. 4A 1-2. Several dashed lines seem to imply hydrophobic interactions (W32-A249, W32-F250, W30-F250, V98-F49), but the selected atoms appear to be rather far apart (4-5 Å). Was any distance cut-off used to select the atoms connected by

dashed lines? Similarly, do the side chains of D88 and R94 interact directly?

The dashed lines in the figure are indeed meant to indicate potential hydrophobic interactions between selected residues. We acknowledge that the distances between these atoms are in the range of 3.5–5 Å and some of them appear to be on the upper limit for hydrophobic interactions.

In our analysis, we employed a standard distance cutoff of 5 Å to identify and depict potential hydrophobic interactions (Lin, Matthew S. et al. PMID: 17562319 Figure 1). This cutoff is commonly used in structural biology to capture significant van der Waals interactions between nonpolar atoms, which contribute to hydrophobic effects within this distance range. While interactions closer than 4 Å are generally more significant, those within the 4–5 Å range can still contribute to the overall stability and conformation of the protein, particularly in dense or complex environments where multiple weak interactions collectively play a role. These interactions, while weaker, are not negligible and are included to provide a more complete picture of the molecular environment. We have amended our figure to colour code the distance between residues, showing those below 4 Å in red, 4.0 to 4.5 Å in orange and 4.5 to 5.0 Å in yellow (**see new Figure 4A and Figure 5E**).

3. Have the authors considered depositing selected AlphaFold models (e.g. with iptm+ptm scores higher than 0.5 and/or the one discussed in more detail in the manuscript — three TGF-beta isoforms plus BG/InhA) as supplementary information files (in PDB format)?

In the revised manuscript we have included AF2M models of both TGF-β2:BG_{ZPC} and TGF-β2:TGFBR1:TGFBR2:BG_O (see revised Supplementary data).

4. Lines 509-513. Given that the authors explicitly discuss pH-dependent interactions between BMPs/GDFs and RGMs, wouldn't it be appropriate to consider citing two articles that demonstrate these pH-dependent interactions between BMP2/GDF5 and RGMs (PMID 25938661 and 32576689)? Full disclosure: this reviewer is the first author of PMID 32576689.

We thank the reviewer for this suggestion and both publications are now included in the manuscript.

5. Line 600. Change RO1 to R01.

Corrected

6. Line 638. Change '30..853' to '30–853'.

Corrected

7. Line 660. In 'rat BGO', should 'O' be in subscript?

Corrected

8. Line 662. Please provide more details on the BG(ZP-C) construct (amino acid residue range, linkers, protease cleavage sites, etc.), its expression (duration, temperature, etc.), and purification (buffers and purification columns). Ideally, please include the full-length amino acid sequence of the construct in the supplementary information file.

We have now added additional details regarding the BG_{ZP-C} construct, including the amino acid residue range, linkers, and protease cleavage sites (**page 24 lines 655-666**). For the protein expression and purification protocols, due to the complexity and length of the procedure, we refer to our recently published paper (Borgini, et al., PMID: 3796466), which provides a comprehensive description of all steps. This publication is available as open access, ensuring that all details are readily accessible to readers.

9. Lines 669-674. Please provide more details on the Inhibin A construct (amino acid residue ranges, secretion signal, etc.), its expression (duration, temperature, etc.) and purification (all buffers and purification columns). Ideally, please include the full-length amino acid sequence of the construct in the supplementary information file.

We have added additional details regarding the InhA construct, including the amino acid residue range, linkers, and protease cleavage sites (**page 24-25 lines 668-676**).

10. Lines 728-756. Why were the studies performed in 10 mM CHES at pH 8.0-8.6? Are BG(O) proteins more soluble or stable in this buffer? If so, it might be useful to indicate this.

We used a pH range of 8.0–8.6 to minimize nonspecific interactions between BG_O and the chip surface. A pH screening was conducted between 6.0 and 9.0, and we found that the values around pH 8.6 provided optimal conditions for reducing these unwanted interactions. We have now included this rationale in the methods section of the manuscript (see **page 27, lines 745-746 and 759-760**).

11. Line 750. What is 'the A-B-A injection scheme'? Please clarify.

The A-B-A injection mode in the Biacore T200 system refers to a specific injection scheme. In this mode, buffer (A) is first injected over the sensor chip surface to establish a baseline. This is followed by the injection of the analyte (B), allowing for the observation of any binding interactions. After the analyte injection, another injection of buffer (A) is performed to monitor the dissociation phase and return the sensor surface to baseline conditions. This sequence ensures that any changes in the response are attributed to the analyte interaction rather than nonspecific effects or baseline drift. We have clarified the description of the A-B-A mode in the methods section (see **page 27-28, lines 762 - 765**).

12. Lines 784-804. How/were crystals cryoprotected? If so, please indicate the concentration of cryoprotectants.

No cryoprotectant was used. Excess solution from the loop was carefully wicked off during crystal handling. The crystals were then immediately flash-cooled in liquid nitrogen. We have added this clarification to the methods section of the manuscript (see **page 29, line 807**).

13. Line 810. Change 'keV' to 'kV'. Also, make this change in lines 834, 857, and 882.

Corrected

14. Lines 813 and 885. Change 'contrast functions' to 'contrast transfer functions'.

Corrected

15. Lines 806-828. Please cite papers describing CryoSPARC (which version was used?) and RELION-4.0.

This has now been done across the Methods section.

16. Line 839. Change 'Particles' to 'particles'.

Corrected

17. Multiple places. Why do certain words start with a capital letter in the middle of a sentence? For example, 'Grids' (line 854) and 'Movies' (lines 857 and 881). It should be RELION-4.0, not Relion 4.0: <https://relion.readthedocs.io/en/release-4.0/>

Corrected

18. Figure 3 legend. Change 'Detailes' to 'Details'.

Corrected

19. Figure 4D legend. Change 'into the binding interface' to 'to the binding interface'? Change 'PISAE PDB' to 'PDBE PISA': <https://www.ebi.ac.uk/pdbe/pisa/>

Corrected

20. Figure 4D legend. 'Growth Factors from TGF-beta family'. Were BMPs/GDFs included in this analysis? If so, shouldn't it be 'Growth Factors from the TGF-beta superfamily'?

In the field the preferred term is now 'TGF- β family' and not 'TGF- β superfamily' (see PMID 27141051), as the ligands are all highly related at the sequence level and also at the structural and functional levels. A superfamily, in contrast, is a group of proteins that have a common ancestor, and show structural homology, but not necessarily much recognisable sequence homology.

21. Sensorgrams in all main figures. Wouldn't it be easier for the reader to see the K_D s in these figures straight away?

We have added the K_D values alongside the presented sensorgrams (see new **Figure 3C, 5A–C, F, G, 6I, Extended Data Figure 3A, 4A, 5E, 6A, B**).

22. Fig. 5E. It is not clear what the yellow dashed lines indicate. Some residues appear to be too far apart to form hydrophobic interactions (e.g., I39 and Y248). It is difficult to determine without further information (was any distance cut-off used?).

As for previous figure, we colour-coded the distance between residues, showing those below 4 Å red, 4.0 to 4.5 Å orange and 4.5 to 5.0 Å yellow.

23. Fig. 6D. Change 'Instensity' to 'Intensity'.

This has now been corrected

24. Fig. 7 legend. Both British and American English are used, such as 'signaling' and 'signalling'.

This has now been corrected

25. PDB ID 8DC0. Water molecules (their oxygen atoms) 62 and 76 clash with neighbouring protein oxygen atoms (distances of 1.96 and 1.91 angstroms, respectively). This likely contributes to the relatively high clashscore (7.88). I used Coot-Validate-Probe clashes. These are the top four clash hits.

In the revised version we have improved the structure factors and now the clash score is significantly lower (see PDB entry 8DC0).

Reviewer #2 (Remarks to the Author):

This work deals with the cell surface glycoprotein betaglycan (BG), and in particular on its interaction with different binding partners, from a structural focus with the aim of understanding its mechanisms of action. Due to my background I'll focus on the NMR experiments.

NMR is used with two main purposes here:

- to demonstrate that the mutations done in different domains of BG (BG<ZP-Csub> and zfBG<osub>) to test their impact, and thus relevance, in protein binding partners interactions, do not perturb BG domain structure.

- to monitor the interaction in solution of BG<ZP-Csub> with three protein ligands: TGF- β 2, mmTGF- β 2, and Inhibin A.

With respect to the first issue:

- in page 11, line 291: "we recorded NMR natural abundance 2D 1H-13C shift correlation": 2D 1H-13C shift correlation is not any type of NMR spectra, it could be better "2D 1H-13C correlation spectra" or "2D 1H-13C HMQC spectra". "... and 1D 1H spectra blabla" should be removed.

We have amended the description accordingly (see **page 11 line 293-294**).

The conclusion of this paragraph should be less categorical: it is true that the NMR fingerprints of the mutants and wild type are very similar, but still there are differences, and we cannot assure that they do not arise from small changes in the structure. It should be better said that the mutations did not cause important changes in the overall protein structure.

We apologise for the overstatement. In the revised manuscript, we have amended the wording to reflect the fact that while the NMR fingerprints of the mutants and wild-type are very similar, there are still minor differences, and we cannot rule out the possibility of small structural changes (see **page 12 lines 321-322**).

- there is some confusion with the mutations at zfBG<osub>: in the manuscript is says N253A but in extended figure 4 it says D253A.

We are grateful to the reviewer for catching this error. We have corrected the description, and it should indeed be D253A throughout the manuscript.

Second issue:

- In Figure 6A-C, ppm of both axes should be added.

We have updated the figure accordingly.

In page 15, line 415: "Upon titration of 13C-methyl-Ile BGZP-C with InhA, we observed a concentration dependent weakening of signal intensities and peak broadening, which is indicative of binding (Fig. 6C-E)." Well, not necessarily, it could just mean BG

aggregation or oligomerization? I would use "suggesting" rather than "indicating". Interestingly, this domain contains 6 Ile, as in fact reflected in the MeTROSY spectrum. Interestingly, all these Ile, except I102 gather in the same region in the 3D structure. Could it be that the InhA binding region is far away from any Ile residue (or in other words, the probe used here, which are Ile, are not good for detecting binding to InhA). In fact, the fact that I161 does not move indicates that the conformational change (from random coil to helix) observed for TGF- β binding does not happen upon InhA binding. It looks indeed that binding is happening, maybe accompanied with some oligomerization process (that's why the broadening)?, maybe at a different region?

We thank the reviewer for this valuable suggestion and agree that this may be the case. This NMR experiment agrees well with the SPR data, where mutations of BG_{ZP-C} that abrogate binding to TGF- β 2 do not significantly impact binding of InhA, suggesting that a different region might be involved. In contrast, from the competition experiment, we see that both ligands compete, suggesting both interfaces must at least partially overlap. From the extended methyl spectra (ILVM) (Figure for Reviewer #2), we see that none of the residues demonstrate significant shifts, but the spectra intensity and line width is affected. Based on this, and the observation (not shown) that ActA, which differs only from InhA in the ZPC-binding alpha subunit with a non-binding β A subunit, did not lead to decreases in signal intensity or peak broadening, we believe that our NMR experiment indicates that binding occurs. However, as the reviewer suggests, oligomerization processes are probably also at play. We have amended the description in the text to reflect these points (see **page 15 lines 418-419**).

Figure for Reviewer #2. ^1H - ^{13}C methyl correlation NMR spectra of $\text{BG}_{\text{ZP-C}}$ (black) titrated with InhA (red). Methyl labelled (ILVM) $\text{BG}_{\text{ZP-C}}$ was titrated with InhA and IBS_SOFAST experiment was recorded on 600Mhz spectrometer. No significant shifts were detected

Reviewer #3 (Remarks to the Author):

From a general perspective, I think that the work is highly interesting and provides new data that allow advancing the current knowledge in TFG recognition and signalling. The methodological approach used is properly designed with the combination of x-ray, cryo EM and NMR.

We are very pleased to see that this reviewer thought our work was highly interesting and considered that it provides new data that will advance the current knowledge of TGF- β recognition and signalling.

First, I would like to clarify that my specific comments are focus just on the NMR experiments.

In this context, some additional information should be provided to clarified certain points:

1) Regarding Extended Figure 4 and page 12 main text, where it is said: "Notably, the D253A mutant exhibits signal broadening across the spectra, hinting at more widespread structural changes that may suggest alterations in protein folding".

And page 12 where it is said: "One caveat to these results is that the N253A mutation appeared to perturb the structure".

We are grateful to the reviewer for catching this error. We have corrected the description, and it should indeed be D253A throughout the manuscript.

The authors should further clarify this data interpretation since the signal broadening across the spectra could be also related to a change in the aggregation state of the protein. Then, further information should be required to assess that the D253A mutant displays a change in protein folding. For instance, a methyl-TROSY 1H-13C could be measured to confirm that there are chemical shift changes that correlate with a different protein folding and not only line broadening in this construct.

We thank the reviewer for this valuable suggestion. We agree that signal broadening could potentially result from changes in the aggregation state of the protein. Based on the 1D proton spectra we recorded, we observed that the D253A mutant shows noticeable differences compared to the wild type, indicating that this variant may indeed be subject to global changes, such as aggregation, rather than the mutation mediating changes in the local environment. Given that the spectra indicate substantial deviations from the wild type, we now mention that the observed behaviour in the SPR experiment may not entirely reflect local changes (see **page 12 line 321-322**). While further detailed investigation (e.g., additional methyl-TROSY 1H-13C measurements) could provide deeper insights into the properties of this particular mutant, we believe the current NMR data provide sufficient information to cautiously interpret the SPR data.

2) In Figure 6 the 1H and 13C scales of the NMR spectra are missing, the scales should be included. Moreover, the type of 1H-13C correlation experiment that has been used should be specify in the figure caption.

We have updated the figure and the description accordingly.

3) Finally, the authors should clarify the assignment protocol that has been carried out to identify the chemical shifts of delta1 methyl of Ile 161.

We have added the assignment protocol for Ile161 (that was conducted by mutagenesis of Ile161 to Ala with subsequent NMR ¹H-¹³C correlation spectra acquisition) to the materials and methods section (see **page 25-26 lines 699-706**).

Reviewer #4 (Remarks to the Author):

The paper by Wieteska and colleagues presents complementary structural approaches (X-ray crystallography, NMR and cryo-EM) towards solving the structure of the co-receptor betaglycan (BG) with its ligands (TGF-beta1, -beta2, beta3, inhibin-A) and further even the high molecular complex of BG-ligands and signaling type II and type I receptors of TGF-beta. The paper comes as a natural sequel after a series of structural papers on BG generated by the AP Hinck lab. The unique contribution of the new paper is the ingenious bypass of several hurdles that prohibit the crystallization of large multi-domain parts of BG together with its ligands and receptors. The combination of clever mutant ligands, multiple ligands and BG from zebrafish and rat take the structural TGF-beta field one step further and allows the authors to present exciting new structural models of these important receptors bound to several of their ligands, thus advancing our understanding of signal transduction and refining older observations based on a solid perspective from the structural field.

The paper is succinctly written and provides a large resource of new structures. I have some conceptual comments that relate to the interpretation of the data and the signalling complex, few comments on missing controls and a general request on presentation of statistical details in the relevant data panels.

We were very pleased to see that the reviewer thought that our paper presents exciting new structural models which will advance our understanding of signal transduction.

Comments:

1. To assist the non-specialist, figure 1 can start with a cartoon of one ligand and the three receptors, aiming at highlighting the relevant domains of each protein that are analyzed in the paper. This is most relevant for the BG domains: BGO, BGZP-N and BGZP-C.

We thank the reviewer for this valuable suggestion. In the revised version of the manuscript, we have included a schematic of betaglycan that highlights the domain composition and arrangement (see **new Figure 1A**). Since panel B (previously panel A) already contains the structure of TGF- β , the cartoon we have added focuses only on the domain structure of betaglycan. This should help readers understand the key aspects of betaglycan analyzed in the paper.

2. Figure 2A: by comparing DAPI and BG staining, not all cells appear to express strong receptor levels in this stable clone of the cells. The TGF-beta2 staining pattern appears to overlap exactly with the BG pattern. Yet, can the fluorescent TGF-beta2 bind to endogenous (unlabeled) receptors? This does not complicate the co-localization analysis but is relevant to the signaling question presented below. Explain in the legend that this is rat BG.

It is correct that not all cells express high levels of SNAP-BG, as we deliberately included a pool of cells with varying SNAP-BG expression levels. This design provides an internal control within every experiment, allowing us to clearly demonstrate that colocalization of the signals is directly dependent on the level of SNAP-BG expression.

While TGF- β 2 can bind to endogenous receptors, the overall expression level of these receptors is substantially lower compared to the overexpressed SNAP-BG, contributing only minimally to the total signal (see response to question 3 below). To further clarify this point, we have included a **new control panel D in Extended Data Figure 2**, which shows that for untransfected HEK293T cells, there is no significant signal from bound TGF- β 2

3. Ext data Figure 2A: the SNAP-BG appears as a sharp band of 120 kDa. Does this BG become GAGylated generating a long smear on the gel? To get a feeling on the degree of overexpression, this western blot could be re-probed with a BG antibody that will also detect the endogenous human BG. Explain in the legend that this is rat BG.

The SNAP-BG construct used in this study is the Δ GAG variant, meaning the higher molecular weight species observed are mostly due to N- or O-glycosylation. However, the dominant population clusters around 120 kDa. To answer the reviewer's question about levels of betaglycan overexpression, we have performed an additional Western blot analysis using an anti-betaglycan (BG) antibody (see **new Extended Data Figure 2A**). As shown, the endogenous levels of BG are extremely low and essentially undetectable by this method.

4. Ext data Figure 2C: for how long were the cells exposed to TGF-beta2? In this experiment, the signaling in the minus SNAP-BG lanes is mediated by the endogenous BG, right? This is why in the previous comment, monitoring the endogenous BG was suggested.

We thank the reviewer for this valuable comment. In this experiment, we followed the protocol outlined by Esparza-López et al. (PMID: 11278442). The optimal signalling response difference between transfected/untransfected cells was visible just after 15 minutes of exposure to TGF- β 2. We used the L6E9 mouse myoblast cell line, which exhibits minimal to no endogenous betaglycan expression (PMID: 8391934). We have clarified the figure description accordingly to avoid any potential confusion.

5. The discussion presents the possible scenario of BG internalization, low pH and dissociation of BG from the ligand-receptor complex. Can the SNAP-BG system also monitor BG internalization together with TGF-beta2? Do the two signals dissociate in acidic endosomes

We thank the reviewer for raising this interesting point. We agree that the SNAP-BG system could potentially be a useful tool for monitoring the internalization of betaglycan (BG) together with TGF- β 2. This is indeed an intriguing possibility, and we plan to explore this in future experiments. However, tracking the co-internalization and potential dissociation of the two signals in acidic endosomes presents considerable technical challenges, and thus will be the subject of future work and is certainly beyond the scope of the current manuscript.

6. Figure 3E indicates similar KD values for TGF-beta2 binding to BGO and BGZP-C, around 100 nM. Figure 4C indicates KD value for TGF-beta2 binding to BGO, around 50 nM. Is it safe to assume that the 2 domains, BGO and BGZP-C, have equal affinity

for the ligand? In the BG dissociation model, does the binding of type I receptor decrease the K_D significantly? I wonder whether the K_D for ligand binding of each domain is different when the domain is present in native BG and not isolated.

We thank the reviewer for bringing this to our attention. We believe the difference in affinity observed between Figures 3E and 4C is primarily due to the different ligand immobilization techniques used in the SPR experiments. In Figure 3E, the sensorgrams were obtained using a Biacore T200 at the Francis Crick Institute, where TGF- β 2, TGF- β 2-YL, and TGF- β 2-QKL were immobilized on a C4 chip via amine coupling, achieving a surface density of approximately 400 RU. In contrast, the SPR experiment shown in Figure 4C was performed on a Biacore X100 at the University of Pittsburgh, where minimally biotinylated TGF- β 2 was immobilized to a surface density of around 50 RU. It is important to note that the primary goal of these analyses was to identify potential hot spots for ligand–receptor interactions, and the differences in affinity due to immobilization techniques are not significant for the conclusions we have drawn from the data.

In response to the second question, BGo is unable to bind TGF- β when TGFBR1 and TGFBR2 form a ternary complex with TGF- β (Villarreal et al. PMID: 27951653). Consistent with this, our structural analysis indicates that BGo and TGFBR1 share part of the same binding interface. Furthermore, evidence shows that TGFBR1 can displace BGo from its position in the TGF- β :(TGFBR2)₂:BGo complex, while even full BG cannot outcompete TGFBR1 (PMID 27951653, Figure 8). We propose therefore that the individual BG domains exhibit similar affinity for TGF- β , whether isolated or within BG, as they are separated by long linkers and do not interact directly (see **Figure 1L and Extended Data Figure 1G**). However, due to avidity, the K_D for BG binding to TGF- β is significantly higher than for each domain individually.

7. Figure 5A-C: is the difference between BG from zebrafish and rat conserved also in human BG?

Yes, indeed the sequence of human BGo domain and rat are almost identical within the TGF- β binding interface (see Figure 1 for Reviewer #4 below). Thus, the difference observed between rat and zebrafish will also be true for human versus zebrafish BGo. Interestingly, the truncated variant found in zebrafish is also present in other members of the bony fish family, but the ancestral BGo for this family is more similar to the rat and human BGo, suggesting a divergent evolution of the BG protein in teleosts.

RAT	4	CELSPINASHPVQALMESFFIVLSGCASRGTGLPREVHVLNLRSTDQGPQQRQREVTLHL	63
		CELSP++ASHPVQALMESFFIVLSGCASRGTGLP+EVHVLNLR+ QGPGQ QREVTLHL	
HUMAN	28	CELSPVSASHPVQALMESFFIVLSGCASRGTGLPQEVHVLNLR+TAGQGPQLQREVTLHL	87
RAT	64	NPIASVHTHHKPIVFLNLSPOPLVWHLKTERLAAGVPRFLVSEGSVVQFPNGNFSLTAE	123
		NPI+SVH HHK +VFLNLS PLVWHLKTERLA GV RFLVSEGSVVQF S NFSLTAE	
HUMAN	88	NPFISSVHIHHKSVVFLNLSPHPLVWHLKTERLATGVSRLFLVSEGSVVQFSSANFSLTAE	147
RAT	124	TEERNFPQENEHLLRWAQKEYGAVTSFTELKIARNIYIKVGEDQVFPPTCNIGNFNLFLSLN	183
		TEERNFP NEHLL WA+KEYGAVTSFTELKIARNIYIKVGEDQVFP CNIGNFNLFLSLN	
HUMAN	148	TEERNFPHGNEHLLNWARKEYGAVTSFTELKIARNIYIKVGEDQVFPKCNIGNFNLFLSLN	207
RAT	184	YLAEYLQPKAAEGCVLPSQPHEKEVHIIELITPSSNPYSAFQVDIIVDIRPAQEDPEVVK	243
		YLAEYLQPKAAEGCV+ SQP +EVHIIELITP+SNPYSAFQVDI +DIRP+QED EVVK	
HUMAN	208	YLAEYLQPKAAEGCVMSQPQNEEVHIIELITPNSNPYSAFQVDITIDIRPSQEDLEEVVK	267
RAT	244	NLVLILKCKKSVNWVIKSFQVKGSLKVIAPNSIGFGKESERSMTMTKLRDDIPSTQENL	303
		NL+LILKCKKSVNWVIKSFQVKG+LK+IAPNSIGFGKESERSMTMTK +RDDIPSTQ NL	
HUMAN	268	NLILILKCKKSVNWVIKSFQVKGSLKIIAPNSIGFGKESERSMTMTKSIKIRDDIPSTQGNL	327
RAT	304	MKWALDNGYRPVTSYTMAPVANRFHLRLEN	333
		+KWALDNGY P+TSYTMAPVANRFHLRLEN	
HUMAN	328	VKWALDNGYSPITSYTMAPVANRFHLRLEN	357

Figure 1 for Reviewer #4. Sequence alignment between human and rat BGO domain. Highlighted in red are the region directly taking part in TGF- β binding

8. The discussion and figure 7 present the model of TGF-beta signaling. This raises some fundamental questions. I recommend that the authors include some further clarifications and a short expansion of the discussion. Why is it not preferable that the complex shown in Figure 1I is the signaling complex? Is such a complex with BG and only one set of type II/type I receptors expected to signal "less" compared to a complex with two sets of II/I receptors? If there is evidence for this, it is worth presenting it in some detail in the discussion.

We thank the reviewer for the interesting question. Indeed, it has been demonstrated that the engineered heterodimer, which recruits signalling receptors only on one side of TGF- β , can still signal, although its potency is reduced compared to the full dimer (Huang, Tao, et al., PMID: 21423151). In that study, pSMAD3 levels were analysed via western blot following stimulation with either a fully functional homodimer or a half-functional heterodimer. The phosphorylation levels for the half-functional heterodimer were significantly lower than for the homodimer but remained substantially higher than the control. Moreover, Villarreal et al. (PMID: 27951653) suggested that both BGO and full BG are replaced by signalling receptors at both binding sites. Based on this reviewer's comments, we have amended the discussion to include these additional details and further clarify this point (see **page 18 lines 509-514**).

9. Continuing in the same line of discussion, the authors discuss the importance of receptor internalization and of receptor cleavage, which is relevant and appropriate. In the same context, it would be useful to discuss also the mechanism of ligand-coreceptor assembly from the latent form of the ligand. It is obvious that all solved BG structures use mature ligands or even mutants like the WD mutant used in this study for technical reasons. A brief discussion on whether the authors envision first the generation of a local pool of mature ligand from its latent depots, that then binds to BG with the calculated KD from the studies in vitro, and then the formation of the hyper-complex analyzed in this paper, will enlighten the reader.

We had indeed considered the possibility that one function of betaglycan might be to facilitate the release of the mature growth factor from the pro-complex. However, after analyzing the structures, our initial enthusiasm for this hypothesis diminished. The mature TGF- β utilizes similar residues to bind both the pro-domain and betaglycan, effectively shielding it from such interactions.

Nevertheless, we attempted to form a complex between pro-TGF- β 2 and either BG_O or full-length betaglycan. As shown below in **new Extended Data Figure 7**, no such complex could be formed. Additionally, we did not observe any "hijacking" of the ligand by either BG_O or betaglycan. While this interaction could not be demonstrated in our experiments, we acknowledge that in a cellular context—where additional factors such as shearing forces may come into play—this possibility cannot be entirely ruled out. We have amended the results and discussion to reflect these points based on the reviewer's suggestion (see **page 16 lines 441-452 and page 19-20 lines 544-553**).

Technical comments:

1. Figure 1C, F, H, K and other figures later: explain in the legend the importance of the 9 microphotographs. They are presented as corresponding 2D classes but the method presents millions or hundreds of thousands of classifications, so what do the 9 images represent, and why 9?

We have updated the figure description accordingly. The 9 selected 2D classes provide a representative overview of the dataset quality, illustrating the coverage of different particle orientations and the redundancy within the data. While hundreds of thousands of classifications are generated, these specific panels offer a visually manageable and informative summary, capturing the key structural features and the diversity of views present in the dataset.

2. Figure 1F, H: explain in the legend the SEC peaks labeled as (b).

The Figure has been corrected. The labelling for the peak representing the partial complex (b) has been removed to make figure clearer.

3. Ext data Figure 2D: explain in the legend what each data point represents. A single cell?

We thank reviewer for catching this lack of clarity. Each data point represents the ratio of SNAP-BG signal to TGF- β 2 signal within a segmented region of the image. Segments were generated by applying the OTSU thresholding method to the SNAP-BG channel, isolating areas of significant fluorescence. We added this information to the relevant figure legends (**Figures 2B, 6E, Extended Data Figures 2E, 6C**).

4. Figure 1B: explain in the legend what each data point represents. A single cell?

Each data point in Figure 2B represents the ratio of SNAP-BG signal to TGF- β 2 signal within a segmented region of the image generated by applying the Otsu thresholding method to the SNAP-BG channel. We have updated the figure legend and the Materials and Methods section accordingly (see **page 35 lines 987 -989**).

5. Figure 1C, D: explain in the legend the statistical parameters: standard deviation or SEM, number of repeats.

This has been corrected.

6. Figure 1C, D and later: include a method on AF2M and explain iptm and ptm.

We have added a detailed description of the AF2M protocol to the Materials and Methods section (see **page 35 lines 990-999**) and updated the figure legend accordingly.

7. Figure 3B, E, F (luciferase) and additional SPR assays in the other figures: explain in the legend the statistical parameters: standard deviation or SEM, number of repeats. In panel 3F, also explain the p-value of 4 stars.

We have updated the figure legend accordingly.

8. Figure 4E: explain in the legend that the numbers on top of the brackets refer to the number of mutations. The quadruple mutant requires a bracket with a 4.

We have updated the figure accordingly.

9. Ext data Figure 5D, E: in the legend, correct the panel labeling. First D and the last panel is E.

We have updated the figure accordingly.

10. Figure 6D, E: explain in the legend the details of these diagrams: median line in panel D, quartiles in panel E?

We have modified the figure and updated the legend accordingly.